# Domain-Agnostic Molecular Generation with Chemical Feedback

Yin Fang♣♠, Ningyu Zhang♣♠*, Zhuo Chen♣♠, Lingbing Guo♣♠, Xiaohui Fan♣, Huajun Chen♣♠♡*

♣ College of Computer Science and Technology, Zhejiang University
♠ ZJU-Ant Group Joint Research Center for Knowledge Graphs, Zhejiang University
♡ ZJU-Hangzhou Global Scientific and Technological Innovation Center, Zhejiang University
{fangyin, zhangningyu, zhuo.chen, lbguo, fanxh, huajunsir}@zju.edu.cn

## Abstract

The generation of molecules with desired properties has become increasingly popular, revolutionizing the way scientists design molecular structures and providing valuable support for chemical and drug design. However, despite the potential of language models in molecule generation, they face challenges such as generating syntactically or chemically flawed molecules, having narrow domain focus, and struggling to create diverse and feasible molecules due to limited annotated data or external molecular databases. To tackle these challenges, we introduce MOL-GEN, a pre-trained molecular language model tailored specifically for molecule generation. Through the reconstruction of over 100 million molecular SELFIES, MOLGEN internalizes structural and grammatical insights. This is further enhanced by domain-agnostic molecular prefix tuning, fostering robust knowledge transfer across diverse domains. Importantly, our chemical feedback paradigm steers the model away from "*molecular hallucinations*", ensuring alignment between the model's estimated probabilities and real-world chemical preferences. Extensive experiments on well-known benchmarks underscore MOLGEN's optimization capabilities in properties such as penalized logP, QED, and molecular docking. Additional analyses confirm its proficiency in accurately capturing molecule distributions, discerning intricate structural patterns, and efficiently exploring the chemical space[1].

## 1 Introduction

Molecule generation – synthesizing and designing novel molecules with desirable properties – holds an important place in chemical science, with numerous applications in drug discovery (Wang et al., 2022). Generating molecules is challenging due to the immense and discrete nature of the molecular space, which, with an estimated size of $10^{33}$, makes exhaustive searches impractical (Polishchuk et al., 2013). Early, deep generative models (Jin et al., 2020; Zang & Wang, 2020; Luo et al., 2021; Shi et al., 2020b) have emerged as one of the most promising tools for exploring the broader synthetically accessible chemical space. These models' ability to automatically generate chemically valid and structurally similar molecules has proven to be invaluable for tasks such as the inverse design of functional compounds (Flam-Shepherd et al., 2022).

Current deep generative models typically involve initial training of an unconditional generative model through a large set of existing molecules, and then use additional reward functions (Cao & Kipf, 2018; Popova et al., 2018; You et al., 2018; Popova et al., 2019; Shi et al., 2020b; Zang & Wang, 2020) or property predictors (Liu et al., 2018; Jin et al., 2019; Gómez-Bombarelli et al., 2018) to guide the synthesis of new molecules with desired properties. However, these approaches are limited by challenges in training due to the high variance of Reinforcement Learning (RL) (Xie et al., 2021), fixed-dimensional latent generation space (Wang et al., 2023), and expert-provided generation rules (Sun et al., 2022), which impede efficient exploration of the broader chemical space.

---

*Corresponding author.
[1]Code is available at https://github.com/zjunlp/MolGen.

Recent advancements in language models have demonstrated great potential for understanding complex molecular distributions (Flam-Shepherd et al., 2022). To gain a more profound comprehension of the underlying molecular structures and their representations, researchers have begun integrating SMILES (Weininger, 1988), a linear string notation for describing molecular structures, with pre-trained language models (PLMs) (Irwin et al., 2022). Despite their widespread use, several issues remain inadequately considered. **Firstly**, the brittleness of SMILES may lead to a high proportion of generated chemically invalid strings, either due to syntactic errors (e.g., not corresponding to molecular graphs) or fundamental chemical principle violations (e.g., exceeding the maximum number of inter-atomic valence bonds) (Krenn et al., 2020). **Secondly**, almost all previous studies have focused primarily on synthetic molecules, neglecting natural products (Du et al., 2022a). Notably, natural products, characterized by enormous scaffold diversity and structural complexity, exhibit a distinct distribution compared to synthetic molecules and confer additional challenges for numerous molecule generation applications such as drug discovery (Atanasov et al., 2021). **Thirdly**, pre-trained molecular language models often succumb to "*molecular hallucinations*". This refers to instances where the generated molecules structurally adhere to chemical rules, yet fail to demonstrate the anticipated chemical activity in practical applications. This occurs because, although the models assimilate a vast array of molecular structural representations during pre-training, yet they might not fully capture the complex relationships with real-world chemistry and biological properties. Some methods attempt to mitigate this issue by using supervised fine-tuning or external databases (Irwin et al., 2022; Wang et al., 2023), but they may constrain the direction of molecular optimization.

To tackle these challenges, we present MOLGEN, a novel pre-trained molecular language model designed for efficient molecule generation. As illustrated in Figure 1, our approach comprises: *(i)* **A two-stage domain-agnostic molecular pre-training.** First, we train bidirectional and auto-regressive Transformers (Vaswani et al., 2017) to reconstruct over 100 million corrupted molecular SELFIES (Krenn et al., 2020). This endows the model with a profound understanding of the structure, grammar, and intrinsic semantic information of SELFIES, an entirely robust molecular language, free from the predicaments of syntactic and semantic inconsistency often associated with conventional SMILES notation. Next, we leverage domain-agnostic molecular prefix tuning, enabling MOLGEN to harness knowledge transferable across diverse domains (i.e., synthetic and natural products), facilitating task adaptation. *(ii)* **A chemical feedback paradigm to alleviate "*molecular hallucinations*".** By aligning the model's generative probabilities with real-world chemical preferences, MOLGEN learns to evaluate and rectify its molecular outputs, ensuring the generation of chemically valid molecules with genuine utility and anticipated properties.

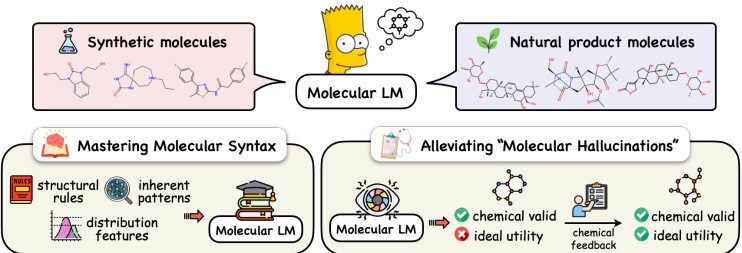

Figure 1: MOLGEN excels at generating chemically valid molecules with expected efficacy in both synthetic and natural product domains.

Through extensive testing on both synthetic and natural product molecular datasets, we establish MOLGEN's capability in producing chemically valid molecules, navigating chemical spaces efficiently, and achieving notable optimization in properties like penalized logp, QED, and molecular docking. Our further analysis underscores MOLGEN's adeptness at understanding complex molecular distributions, recognizing meaningful substructures, and the efficacy of the chemical feedback mechanism, offering novel perspectives and tools to the molecular generation community.

## 2 METHODOLOGY

Figure 2 illustrates the general framework of MOLGEN. The pre-training process (§2.1) comprises two stages: molecular language syntax learning and domain-agnostic molecular prefix tuning. Then, a chemical feedback paradigm (§2.2) is introduced to align the PLM with the anticipated chemical preferences in the downstream phase.

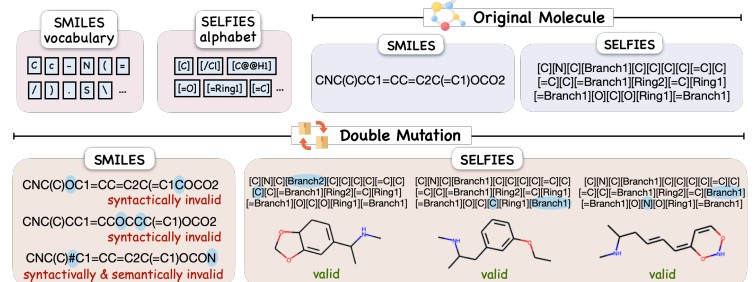

Figure 2: Overview of MOLGEN: pre-training (left) and downstream (right) stages.

## 2.1 DOMAIN-AGNOSTIC MOLECULAR PRE-TRAINING

SMILES and SELFIES are two molecular languages that associate a token sequence with a molecular structure. SMILES denotes molecules as chains of atoms, encapsulating branches within parentheses and signifying ring closures with corresponding number pairs. Despite its longstanding prominence in cheminformatics, SMILES is fundamentally flawed in

Figure 3: Random double mutations of SMILES and SELFIES derived from the same molecule, with blue markings indicating mutation locations. The likelihood of retaining a valid SMILES after a single mutation is 9.9%. For SELFIES, it's a consistent 100% (Krenn et al., 2020).

that it lacks a mechanism to ensure the validity of molecular strings in terms of syntax and physical principles (Krenn et al., 2020). Hence, we employ SELFIES (Krenn et al., 2022), a fully robust molecular language that guarantees every possible combination of symbols in the alphabet corresponds to a chemically sound graph structure. In contrast to SMILES, SELFIES overcomes syntactic invalidity by mapping each token to a specific structure or reference, effectively resolving issues such as unbalanced parentheses or ring identifiers, as depicted in Figure 3. MOLGEN boasts a compact and specialized vocabulary size of 185. While modest in size, this vocabulary is already sufficient to ensure that the language model learns meaningful representations (Rives et al., 2021).

Being the first of its kind to train language models utilizing SELFIES, our work necessitates a solid foundation for comprehending both the syntax and semantics of this language. To achieve a high-quality initialization for MOLGEN, we employ BART model (Lewis et al., 2020) during **the first stage of pre-training**, as shown in Figure 2. Firstly, we convert 100 million unlabeled molecules into SELFIES strings. The standardized representation of SELFIES facilitates the direct construction of an alphabet from the dataset, eliminating the need for a separate tokenizer to discern frequent substrings, thereby preventing the generation of nonsensical tokens. Secondly, we randomly select tokens from the original SELFIES string $S = \{s_1, \cdots, s_j, \cdots, s_l\}$ and replace them with a special token $[\text{MASK}]$. Finally, we encode the corrupted SELFIES using a bidirectional model and calculate the likelihood of $S$ with a left-to-right autoregressive decoder. Formally, the cross-entropy between the decoder's output and the original input constitutes the reconstruction loss:

$$\mathcal{L}_{\text{ce}}(S) = -\sum_{j=1}^{l} \sum_{s} p_{\text{true}}\left(s|S, S_{<j}\right) \log p_\theta\left(s|S, S_{<j}; \theta\right), \quad (1)$$

where $S_{<j}$ denotes the partitial original sequence $\{s_0, \cdots, s_{j-1}\}$, $s_0$ is a pre-defined start token . $p_{\text{true}}$ refers to the one-hot distribution obtained under the standard maximum likelihood estimation:

$$p_{\text{true}}\left(s|S, S_{<j}\right) = \begin{cases} 1, & s = s_j \\ 0, & s \neq s_j \end{cases}. \quad (2)$$

Upon mastering the fundamental grammatical knowledge of SELFIES, we proceed to the second stage of pre-training, wherein we introduce the domain-agnostic molecular prefix as a domain instructor to

facilitate the transfer of knowledge across diverse domains. Unlike the conventional prefix-tuning approach, which exclusively updates the prefix matrices without altering the pre-trained model parameters (Mao et al., 2022; Li & Liang, 2021; He et al., 2022), we capitalize on its influence over the entire model's parameters to effectively bolster its ability to comprehend various domains.

We commence by prepending two sets of $m$ tunable prefix vectors $\boldsymbol{P}_k, \boldsymbol{P}_v \in \mathbb{R}^{m \times d}$, shared among domains, to the keys and values of the multi-head attention at each layer. The output attention score for each head can be formulated as:

$$\text{head} = \text{Attn}\left(\boldsymbol{x}\boldsymbol{W}_q, [\boldsymbol{P}_k,\ \boldsymbol{X}\boldsymbol{W}_k], [\boldsymbol{P}_v,\ \boldsymbol{X}\boldsymbol{W}_v]\right),\tag{3}$$

where $\boldsymbol{X} \in \mathbb{R}^{m \times d}$ denotes the input to a Transformer layer with length $m$, $\boldsymbol{W}_q, \boldsymbol{W}_k, \boldsymbol{W}_v \in \mathbb{R}^{d \times d_h}$ are project matrices that map inputs to queries, keys, and values, and $\boldsymbol{x} \in \mathbf{R}^d$ is a query vector.

Alternatively, the attention between $\boldsymbol{x}$ and $\boldsymbol{X}$ on head can be expressed as:

$$\begin{aligned}
\text{head} &= \text{softmax}\left(\boldsymbol{x}\boldsymbol{W}_q[\boldsymbol{P}_k,\ \boldsymbol{X}\boldsymbol{W}_k]^\top\right)\begin{bmatrix}\boldsymbol{P}_v \\ \boldsymbol{X}\boldsymbol{W}_v\end{bmatrix} = \text{softmax}\left(\boldsymbol{x}\boldsymbol{W}_q\begin{bmatrix}\boldsymbol{P}_k^\top \\ (\boldsymbol{W}_k)^\top(\boldsymbol{X})^\top\end{bmatrix}\right)\begin{bmatrix}\boldsymbol{P}_v \\ \boldsymbol{X}\boldsymbol{W}_v\end{bmatrix} \\
&= \lambda(\boldsymbol{x})\,\text{softmax}\left(\boldsymbol{x}\boldsymbol{W}_q\boldsymbol{P}_k^\top\right)\boldsymbol{P}_v + (1 - \lambda(\boldsymbol{x}))\,\text{softmax}\left(\boldsymbol{x}\boldsymbol{W}_q(\boldsymbol{W}_k)^\top(\boldsymbol{X})^\top\right)\boldsymbol{X}\boldsymbol{W}_v \\
&= \lambda(\boldsymbol{x})\underbrace{\text{Attn}\left(\boldsymbol{x}\boldsymbol{W}_q, \boldsymbol{P}_k, \boldsymbol{P}_v\right)}_{\text{attention of domain-agnostic molecular prefix}} + (1 - \lambda(\boldsymbol{x}))\underbrace{\text{Attn}\left(\boldsymbol{x}\boldsymbol{W}_q, \boldsymbol{X}\boldsymbol{W}_k, \boldsymbol{X}\boldsymbol{W}_v\right)}_{\text{standard attention}},
\end{aligned}$$

$$\tag{4}$$

where $\lambda(\boldsymbol{x})$ is a scalar representing the sum of normalized attention weights on the prefixes.

In this way, domain-agnostic molecular prefixes integrate domain knowledge into the original head attention through linear interpolation. These prefixes are trained simultaneously on different molecular domains, acting as a domain instructor that influences the parameters of the entire model, thereby enhancing the model's mastery of different molecular structural complexities and scaffold diversities.

## 2.2 CHEMICAL FEEDBACK PARADIGM: ALIGN PLM WITH CHEMICAL PREFERENCE

After the pre-training stage, the model gains the capability to generate syntactically correct molecules. However, it may still suffer from "*molecular hallucination*". Consider a scenario where the model is employed to design novel drug molecules. It suggests a molecule with a unique cyclic structure, known to effectively bind with certain drug targets. In an attempt to boost structural robustness, the model introduces an additional side chain. However, this addition, despite seemingly increasing stability, actually interferes with the molecule's intended target interaction, leading to its ineffectiveness. This situation exemplifies "*molecular hallucination*", where the structural enhancements made by the model do not translate into functional success.

**Definition 1.** *Molecular hallucinations* *refer to molecules generated by language models that comply with chemical structural rules, yet fail to exhibit practical utility or the anticipated properties.*

Such hallucinations can hinder drug discovery efficiency, escalate costs, and compromise the real-world applicability of the model. Moreover, an abundance of hallucinated molecules may overshadow truly promising molecular structures. To alleviate "*molecular hallucinations*", we propose a strategy that can effectively gauge and rectify the quality of generated molecular structures. This chemical feedback paradigm ensures that produced molecules are not only syntactically correct but also of high practical utility. Specifically, as illustrated in Figure 2, we align the model's probabilistic rankings of diverse molecular responses with preference rankings observed in actual chemical contexts.

The measure of anticipated chemical preference, denoted as $\text{Ps}(\cdot)$, can be characterized in various ways; in this study, we define it based on the property score. Given a molecule $S = \{s_1, \cdots, s_l\}$, we can generate a set of candidate SELFIES $\mathcal{S}^*$ with distinct property scores using our pre-trained molecular language model. For each $(S_i, S_j)$ pair in $\mathcal{S}^*$ that satisfies $\text{Ps}(S_i) > \text{Ps}(S_j)$, we expect:

$$p_{\text{true}}(S_i|S) > p_{\text{true}}(S_j|S), \quad \forall S_i, S_j \in \mathcal{S}^*,\ \text{Ps}(S_i) > \text{Ps}(S_j).\tag{5}$$

To incentivize the model to assign higher probabilities to candidates with desired properties, we utilize a rank loss (Liu et al., 2022). The rank loss arises when candidates with suboptimal properties obtain higher estimated probabilities compared to those with commendable properties:

$$\mathcal{L}_{\text{rank}}(S) = \sum_i \sum_{j>i} \max\left(0, f\left(S_j\right) - f\left(S_i\right) + \gamma_{ij}\right), \quad \forall i < j,\ \text{Ps}(S_i) > \text{Ps}(S_j),\tag{6}$$

where $\gamma_{ij} = (j - i) * \gamma$ represents the margin multiplied by the difference in rank between the candidates, and $f(S) = \sum_{t=1}^{l} \log p_\theta (s_t \mid S, S_{<t}; \theta)$ denotes the estimated log-probability provided by our pre-trained model with parameters $\theta$. Consequently, we furnish chemical feedback to align the pre-trained model with the chemical preference, without necessitating any supplementary reference data. Unlike supervised fine-tuning, which may still be susceptible to hallucinations due to its reliance on ideal samples, chemical feedback equips the model with a broader perspective. It educates the model on both the commendable and the suboptimal, leading to more informed generation.

Nonetheless, fine-tuning the model solely with sequence-level coordination may diminish its generative capability. To ensure the model retains its generative prowess while optimizing for desired properties, we strike a balance by merging the sequence-level rank loss with token-level cross-entropy loss. The overall loss function is formulated as follows:

$$\mathcal{L} = \mathcal{L}_{ce} + \alpha \mathcal{L}_{rank}, \tag{7}$$

where $\alpha$ is the weight of the rank loss. In practice, we leverage label smoothing (Szegedy et al., 2016) to transform the target distribution $p_{true}$ (Eq. 2) in $\mathcal{L}_{ce}$ (Eq. 1) to a "soft" label, allocating probability mass $\beta$ to other tokens in the alphabet of length $N$:

$$p_{true}(s|S, S_{<j}) = \begin{cases} 1 - \beta, & s = s_j \\ \frac{\beta}{N-1}, & s \neq s_j \end{cases}. \tag{8}$$

Overall, the cross-entropy loss serves as a normalization, complementing the rank loss by ensuring that the model allocates a balanced probability mass throughout the sequence. MOLGEN autonomously steer its learning and optimization paths based on the evaluations of molecules it generates. This cycle of generation and adjustment within the model epitomizes a self-reflective system, even as it incorporates an external scoring function to refine and validate its assessments.

## 3 EXPERIMENTS

### 3.1 EXPERIMENTAL SETUP

In the first stage of pre-training, we randomly select over 100 million unlabelled molecules from the publicly available ZINC-15 dataset (Sterling & Irwin, 2015), which is the same corpus used in Irwin et al. (2022). The chosen molecules meet specific criteria: they're reactive, available for purchase, have a molecular weight of $\leq 500$ Daltons, and a LogP (octanol-water partition coefficient) of $\leq 5$. The second stage includes 2.22 million molecules spanning both synthetic (Irwin et al., 2012; Polykovskiy et al., 2018) and natural product domains (Zhao et al., 2023). In the downstream tasks, as detailed in the following section, we thoroughly investigate the model's capabilities from two perspectives. More information on dataset and experimental procedures are in Appendices C and G.

### 3.2 MAIN RESULTS

#### 3.2.1 MOLGEN CAPTURES REAL-WORLD MOLECULAR DISTRIBUTIONS

An essential capability for any molecular generation model is to capture the molecular distribution and generate diverse and realistic molecules. Such capabilities are paramount when constructing virtual libraries to advance computer-aided drug discovery endeavors (van Hilten et al., 2019). By leveraging a set of compounds, either manually or automatically selected, these models are designed to expand datasets significantly, all the while retaining the implicit structural and chemical patterns inherent to the reference set. In this section, we use seven well-established metrics, detailed in Appendix G, to evaluate the proficiency of models in generating molecules that conform to the distribution of real-world molecules. We generate 10,000 synthetic molecules following the setting in Polykovskiy et al. (2018), and 80,000 natural product molecules based on the pre-trained MOLGEN.

Table 1 reveals the following observations: *(i)* MOLGEN demonstrates a remarkable ability to produce valid molecules without the need for additional valency checks, as required by JT-VAE (Jin et al., 2018). Since LIMO (Eckmann et al., 2022) also employs SELFIES, the generated molecules maintain 100% validity. However, the inherent complexity of natural product scaffolds presents a significant challenge for most models, resulting in a failure to produce valid molecules. The better performance of Chemformer (Irwin et al., 2022) can be attributed to its proficiency in learning SMILES grammar

Table 1: Molecular distribution learning performance on two molecule domains. The cells in highlight denote the best results garnered by MOLGEN and the peak performance achieved by the baselines.

| MODEL | SYNTHETIC MOLECULES | | | | | | | NATURAL PRODUCT MOLECULES | | | | | | |
|---|---|---|---|---|---|---|---|---|---|---|---|---|---|---|
| | Validity↑ | Frag↑ | Scaf↑ | SNN↑ | IntDiv↑ | FCD↓ | Novelty↑ | Validity↑ | Frag↑ | Scaf↑ | SNN↑ | IntDiv↑ | FCD↓ | Novelty↑ |
| AAE | .9368 | .9910 | .9022 | .6081 | .8557 | .5555 | .7931 | .0082 | .9687 | .2638 | .3680 | .8704 | 4.109 | .9943 |
| LATENTGAN | .8966 | .9986 | .8867 | .5132 | .8565 | .2968 | .9498 | .9225 | .2771 | .0884 | .5321 | .6009 | 45.53 | .9949 |
| CHARRNN | .9748 | .9998 | .9242 | .6015 | .8562 | .0732 | .8419 | .7351 | .8816 | .5212 | .4179 | .8756 | 2.212 | .9792 |
| VAE | .9767 | .9994 | .9386 | .6257 | .8558 | .0990 | .6949 | .2627 | .8840 | .4563 | .3950 | .8719 | 4.318 | .9912 |
| JT-VAE | 1.000 | .9965 | .8964 | .5477 | .8551 | .3954 | .9143 | 1.000 | .8798 | .5012 | .3748 | .8743 | 12.03 | .9957 |
| LIMO | 1.000 | .9562 | .1073 | .6125 | .8544 | .1532 | .8956 | 1.000 | .7242 | .0005 | .3416 | .7726 | 31.84 | .9962 |
| CHEMFORMER | .9843 | .9889 | .9248 | .5622 | .8553 | .0061 | .9581 | .9825 | .9826 | .4126 | .5875 | .8650 | .8346 | .9947 |
| MOLGEN | 1.000 | .9999 | .9999 | .9996 | .8567 | .0015 | 1.000 | 1.000 | .9994 | .8404 | .8148 | .8878 | .6519 | .9987 |

during large-scale pre-training, highlighting the importance of pre-training. *(ii)* For the synthetic datasets, most models generate molecules with comparable fragments (Frag) and scaffolds (Scaf) to those of the reference molecules. MOLGEN excels at capturing substructure distributions in natural products, outperforming other models. *(iii)* MOLGEN exhibits the highest SNN and lowest FCD scores, indicating its excellent ability to master the dataset statistics in terms of both biological properties and topological structures. Moreover, its strong performance in IntDiv and Novelty metrics suggests that MOLGEN is well-suited for discovering new chemical structures and exploring unknown chemical space without overfitting. A visual comparison of the training set and generated molecules is presented in Appendix H.1.

### 3.2.2 MOLGEN MITIGATES MOLECULAR HALLUCINATIONS

Addressing the issue of "*molecular hallucinations*" has been a long-standing challenge in the realm of computer-aided molecular design. In this section, we delve into the prowess of MOLGEN in tackling this challenge and primarily focus on two types of experiments: **targeted molecule discovery** and **constrained molecular optimization**. Unlike the molecular distribution learning task, where we only rely on the pre-trained model, here we incorporate the chemical feedback paradigm to align the model with genuine chemical preferences. Specifically, we adopt the penalized logP (p-logP) (Jin et al., 2018), QED (Bickerton et al., 2012) and binding affinity to two protein targets as our optimization criteria, as detailed in Appendix G.

Table 2: Comparison of **QED and penalized logP maximization** methods on synthetic molecules. ♣ indicates output length limit (maximum molecule length of ZINC250K), while ♡ means no limit. The first row summarizes the top 3 property scores from the ZINC250K dataset.

| | MODEL | PENALIZED LOGP | | | QED | | |
|---|---|---|---|---|---|---|---|
| | | 1st | 2nd | 3rd | 1st | 2nd | 3rd |
| | ZINC250K | 4.52 | 4.30 | 4.23 | 0.948 | 0.948 | 0.948 |
| | GCPN | 7.98 | 7.85 | 7.80 | 0.948 | 0.947 | 0.946 |
| | MOLDQN | 11.80 | 11.80 | 11.80 | 0.948 | 0.943 | 0.943 |
| ♣ | LIMO | 10.50 | 9.69 | 9.60 | 0.947 | 0.946 | 0.945 |
| | MOLGEN | 30.51 | 28.98 | 28.95 | 0.948 | 0.948 | 0.948 |
| | JT-VAE | 5.30 | 4.93 | 4.49 | 0.925 | 0.911 | 0.910 |
| | GRAPHAF | 12.23 | 11.29 | 11.05 | 0.948 | 0.948 | 0.947 |
| ♡ | GRAPHDF | 13.70 | 13.18 | 13.17 | 0.948 | 0.948 | 0.948 |
| | MARS | 44.99 | 44.32 | 43.81 | 0.948 | 0.948 | 0.948 |
| | MOLGEN | 80.30 | 74.70 | 69.85 | 0.948 | 0.948 | 0.948 |

Table 3: The top 3 **highest binding affinities** (i.e., lowest dissociation constants, $K_D$, as estimated with AutoDockGPU (Santos-Martins et al., 2021)) from a total of 10k generated molecules for each method.

| MODEL | ESR1 | | | ACAA1 | | |
|---|---|---|---|---|---|---|
| | 1st | 2nd | 3rd | 1st | 2nd | 3rd |
| GCPN | 6.4 | 6.6 | 8.5 | 75 | 83 | 84 |
| MOLDQN | 373 | 588 | 1062 | 240 | 337 | 608 |
| GRAPHDF | 25 | 47 | 51 | 370 | 520 | 590 |
| MARS | 17 | 64 | 69 | 163 | 203 | 236 |
| LIMO | 0.72 | 0.89 | 1.4 | 37 | 37 | 41 |
| MOLGEN | 0.13 | 0.35 | 0.47 | 3.36 | 3.98 | 8.50 |

**Targeted molecule discovery** focuses on generating novel molecules with superior chemical properties. To evaluate model effectiveness, we first present the top-3 property scores of molecules generated on the synthetic dataset in Table 2, following conventions from prior studies (Shi et al., 2020b; Eckmann et al., 2022). It's essential to note that the p-logP score tends to increase linearly with molecule length (Xie et al., 2021; Eckmann et al., 2022). To ensure a fair comparison, we categorize the baselines into two groups. MOLGEN, due to its ability to handle variable-length output, is evaluated under both configurations.

In Table 2, MOLGEN outperforms all baselines in p-logP score and achieves comparable results for QED, indicating the effectiveness of the chemical feedback paradigm in promoting desired molecule probabilities. Further evidence of MOLGEN's capabilities can be found in the results for natural products in Appendix H.2. Given that a mere 0.701% of molecules in our reference set achieve a QED score above 0.9 (with a peak score of 0.9439, as detailed in Appendix C), MOLGEN's achievement of a 0.9478 score highlights its potential in drug discovery. Moreover, the model's ability to produce molecules with a p-logP score of 54.33, substantially exceeding the reference set's high of 17.69.

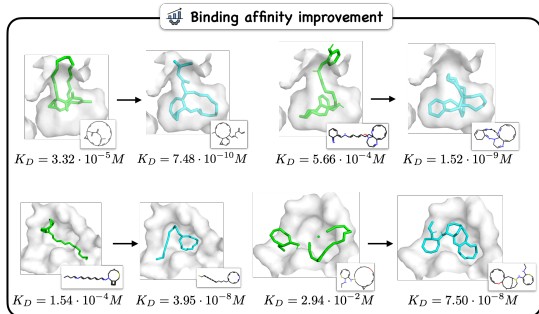

Figure 4: Optimizing ligand binding affinity using MOLGEN. (a) 3D visualization of ligands with the **highest binding affinities** docked against ESR1 (top row) and ACAA1 (bottom row). The protein pocket is displayed semi-opaquely, and the 2D molecular structure of the ligand is shown in the bottom right corner. (b) Examples of **binding affinity improvement** for protein targets ESR1 (top row) and ACAA1 (bottom row).

Moving beyond basic properties, we tackle a more realistic challenge: generating molecules with high binding affinity towards target proteins. Binding affinity quantifies the potency of interactions between a molecule and its intended protein target. Our investigations primarily target the binding sites of two human proteins: the estrogen receptor (PDB ESR1, UniProt P03372) and the peroxisomal acetyl-CoA acyl transferase 1 (PDB ACAA1, UniProt P09110). A detailed exploration of these proteins is available in Appendix G. As shown in Table 3, MOLGEN surpasses prior methods in enhancing binding affinities. Figure 4 (a) illustrates exemplary optimal ligands. To delve deeper into MOLGEN's optimization capability, we undertook an optimization for the 1,000 molecules with the lowest affinities for each protein receptor. Figure 4 (b) offers a comparative visualization of affinity advancements pre- and post-optimization, achieving overall relative improvements of 96.7% for ESR1 and 70.4% for ACAA1. These results illuminate MOLGEN's versatility in both targeted optimization of simpler properties and the more complex domain of molecular docking.

Table 4: Mean (and standard deviation) penalized logP improvement of generated molecules compared to inputs with different similarity constraints.

| MODEL | IMPROVEMENT | |
| --- | --- | --- |
| | $\delta = 0.6$ | $\delta = 0.4$ |
| JT-VAE | 0.28 (0.79) | 1.03 (1.39) |
| GCPN | 0.79 (0.63) | 2.49 (1.30) |
| MOLDQN | 1.86 (1.21) | 3.37 (1.62) |
| VSEQ2SEQ | 2.33 (1.17) | 3.37 (1.75) |
| VJTNN | 2.33 (1.24) | 3.55 (1.67) |
| GA | 3.44 (1.09) | 5.93 (1.41) |
| GRAPHAF | 4.98 (6.49) | 8.21 (6.51) |
| GRAPHDF | 4.51 (5.80) | 9.19 (6.43) |
| LIMO | 1.80 (2.00) | 3.60 (2.30) |
| CHEMFORMER | 2.48 (0.89) | 3.56 (1.32) |
| RETMOL | 3.78 (3.29) | 11.55 (11.27) |
| RT | 2.21 (1.30) | 3.16 (1.50) |
| MOLGEN | **12.08** (0.82) | **12.35** (1.21) |

**Constrained molecular optimization** aims to modify a given molecule to improve desired properties while satisfying a similarity constraint (denoted as $\delta$). Following previous studies (Jin et al., 2018; Shi et al., 2020b; Luo et al., 2021; Eckmann et al., 2022), we optimize 800 molecules from the ZINC250K dataset that exhibit the lowest p-logP scores. To assess the similarity between the optimized and original molecules, we utilize the Tanimoto similarity with Morgan fingerprints (Rogers & Hahn, 2010).

In Table 4, MOLGEN yields superior results under both similarity constraints, illustrating its prowess in scouring the proximate chemical space for molecules with higher property scores. MOLGEN's performance, surpassing models that employ additional reward functions, property predictors, and retrieval databases, confirms that equipping the model with the ability to discern chemical preference is instrumental in alleviating "*molecular hallucinations*".

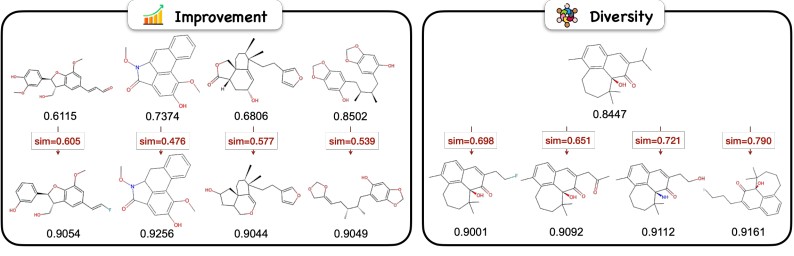

Figure 5: Illustrations of constrained optimization based on QED score within the natural products.

To further probe MOLGEN's capabilities, we expand our constrained optimization experiments to include QED scores for synthetic molecules and both properties for natural products. Figure 5 showcases examples of QED score optimization on natural products. These instances reveal that despite the complex molecular structure and elongated length of natural products, MOLGEN can elevate the property score whilst sustaining a degree of similarity between the input and the modified molecule. Moreover, MOLGEN preserves the diversity

of the generated molecules as it explores the nearby chemical space. Additional visual validations are provided in Appendix H.3.

### 3.3 A Closer Look at MolGen

To dissect the potential of MolGen, we devise experiments from different perspectives.

#### 3.3.1 Pre-training Stage Captures Complex Molecular Characteristics

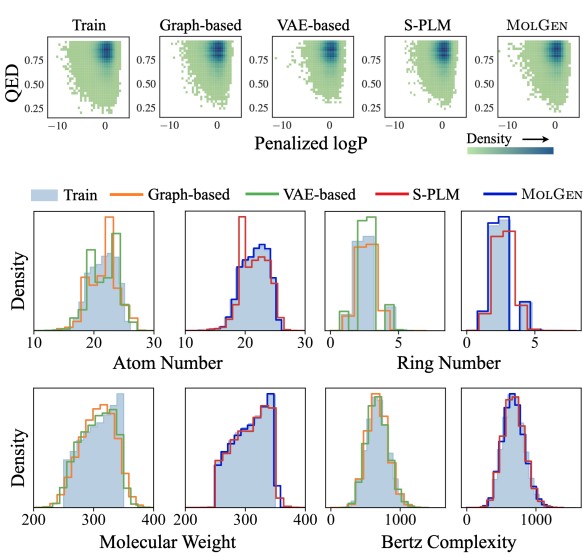

Figure 6: Comparative analysis of molecules generated by different models with respect to properties, atom counts, ring counts, molecular weights, and Bertz complexity (S-PLM stands for SMILES-based PLM).

To understand the differences in property distributions and molecular structures learned during the pre-training phase, we compare the pre-trained MolGen with the most popular deep generative Graph-based (Jin et al., 2018), VAE-based (Blaschke et al., 2018), and SMILES-based language models (Irwin et al., 2022). For this assessment, the training and generation configurations of all models align with the molecular distribution learning task on the synthetic MOSES dataset.

As shown in the 2D histograms of p-logP and QED scores in Figure 6, both VAE-based and SMILES-based PLMs tend to produce molecules with larger p-logP and QED scores than the training data. In comparison, the Graph-based model learns the main mode of p-logP in the training data, while MolGen exhibits a slightly superior performance - analogous outcomes are observed for QED. Furthermore, in terms of molecular topology, PLMs outperform others in perceiving atom numbers, ring numbers, and molecular weights, with MolGen producing a slightly closer match to the training distribution. All the models are proficient at picking up on molecular Bertz complexity. PLMs, particularly MolGen, demonstrate the capacity to capture the properties and structural attributes of the training molecules while maintaining generational diversity.

#### 3.3.2 Chemical Feedback Paradigm Facilitates Property Optimization

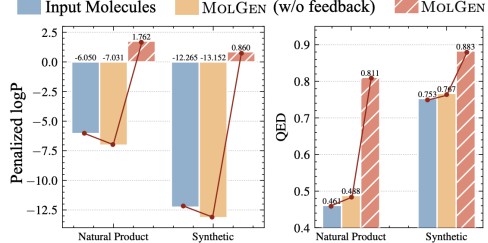
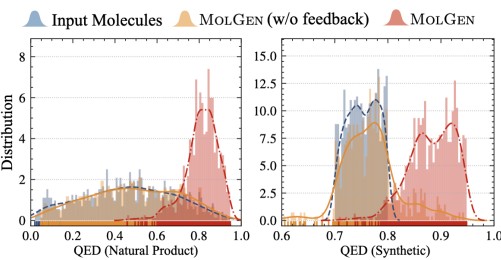

Figure 7: Property variations across different MolGen configurations.

As part of our investigation, we conduct an ablation study to examine the role of the chemical feedback paradigm in mitigating "*molecular hallucinations*". Starting from a batch of molecules from the domains of natural products and synthetic compounds, Figure 7 portrays the variations in property scores of molecules generated by different model configurations. A more comprehensive view of these variations is provided in Appendix H.2.

Without the chemical feedback, the PLM tends to generate molecules with property scores closely resembling those of the initial molecules. This can be attributed to the absence of a guiding signal, leaving the model to rely heavily on its learned patterns from the training data. However, once the

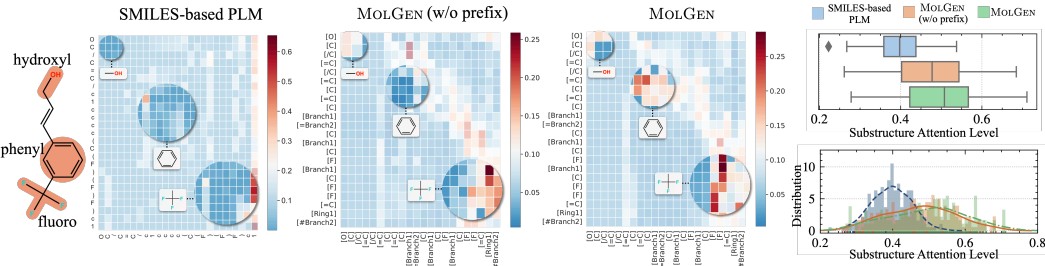

Figure 8: Meaningful substructure attention exploration. Visualization of attention maps learned by various PLM models for the same molecule (left), and substructure attention level of the three models (right). All models being compared are of a similar parameter scale for consistency.

chemical feedback mechanism is integrated, we witness an increase in property scores from the initial to the concluding groups. This underscores the pivotal role of chemical feedback: it furnishes the model with immediate feedback on its performance in generating molecules with the chemical preference, thus steering its outputs towards the desired objectives and alleviating the hallucinations.

### 3.3.3 MOLGEN IMPLICITLY COMPREHENDS MOLECULAR SUBSTRUCTURES

In this section, we investigate PLMs' ability to implicitly discern essential substructures when leveraging different molecular languages (SMILES and SELFIES). For a more intuitive comprehension, we visualize the attention weights of each token within an identical molecule. Specifically, we extract and normalize the attention weights from the final self-attention layer, as depicted in Figure 8.

The attention map generated by MOLGEN shows that the *fluoro* group garners the highest attention weights, followed by the *phenyl* and *hydroxyl* groups. This stems from the *fluoro* group's exceptional electron-capturing capabilities, significantly influencing the molecule's polarity. Meanwhile, the *phenyl* group constitutes a common organic functional group, and the *hydroxyl* group substantially impacts the intermolecular force between the molecule and water. Leveraging domain-agnostic molecular prefixes, MOLGEN directs its attention more efficiently towards these pivotal substructures. These prefixes, acting as domain instructors, enhance the model's adaptability across diverse molecular domains, steering attention away from less essential substructures. Conversely, SMILES-based PLM might divert attention to symbols or numbers devoid of intrinsic chemical significance. Evidently, by employing a precise vocabulary free from such distractions, MOLGEN maintains a clear and implicit understanding of molecular substructures. Further visualizations and analyses supporting this observation are available in Appendix F and  H.4.

To objectively measure the model's focus on vital substructures, we propose a metric termed "Substructure Attention Level (SAL)". This metric is determined by the percentage of attention scores allocated to meaningful substructure tokens within a molecule. Higher SAL scores indicate a stronger focus on meaningful substructures. For effective evaluation, we intentionally select 200 molecules from PubChem, characterized by their simpler structures containing only 1-2 functional groups. This selection criterion ensures that the model's attention isn't diluted across excessively intricate structures, allowing for a clearer reflection of its focus on specific functional groups. The box and distribution plots in Figure 8 vividly depict the SAL of the three PLMs. In line with visualization results, both versions of MolGen surpass the SMILES-based PLM, underscoring MolGen's superior concentration on meaningful substructures. The prefix-enhanced MolGen exhibits a slight edge, highlighting the prefix's role in enhancing attentiveness.

## 4 CONCLUSION AND FUTURE WORK

In this work, we propose MOLGEN, a pre-trained molecular language model specifically tailored for molecule generation. Our in-depth study on MOLGEN confirms its proficiency in generating molecules with chemical preferences while avoiding "*molecular hallucinations*". Furthermore, our model shows potential in identifying essential molecular substructures. Interesting future directions include: *i)* applying MOLGEN to other tasks such as retrosynthesis and reaction prediction (Shi et al., 2020a), *ii)* exploring multimodal pre-training like Edwards et al. (2022); Su et al. (2022); Fang et al. (2024), *iii)* incorporating additional sources of knowledge. We make our pre-trained model, code, and data publicly available, in the hope that our work will foster future research in the field.

## ACKNOWLEDGMENTS

We would like to express gratitude to the anonymous reviewers for kind comments. This work was supported by the National Natural Science Foundation of China (No. 62206246), the Fundamental Research Funds for the Central Universities (226-2023-00138), Zhejiang Provincial Natural Science Foundation of China (No. LGG22F030011), Ningbo Natural Science Foundation (2021J190), CAAI-Huawei MindSpore Open Fund, Yongjiang Talent Introduction Programme (2021A-156-G), CCF-Baidu Open Fund, and Information Technology Center and State Key Lab of CAD&CG, Zhejiang University.

## REPRODUCIBILITY STATEMENT

All data, code, and model weights can be found in the Supplementary Materials. For a detailed description of the dataset, please refer to Appendix C. For specific experimental settings, please see Appendix G.

## ETHICS STATEMENT

This study was carried out in strict accordance with ethical guidelines and best practices in research. The data utilized were sourced from publicly available datasets, and no proprietary or confidential data were used. This study does not involve any ethical issues.

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

## A   AVAILABILITY OF MOLGEN

We have made MOLGEN accessible via **Hugging Face** in support of the broader scientific community[2,3,4]. It is noteworthy that MOLGEN is versatile enough to be applied to tasks beyond the three primary ones discussed in this paper, such as reaction prediction and retrosynthetic analysis. However, due to computational resource constraints, our experimentation is confined to the generation tasks within this study.

It's important to note that our generation task is different from 3D molecular generation. In 3D molecular generation, methods usually consider spatial conformations, bond angles, bond lengths, and other three-dimensional structural aspects of molecules. These approaches often use molecular force fields and molecular docking techniques to optimize the three-dimensional structures of generated molecules. In contrast, 2D molecular generation aims to create two-dimensional flat structures that capture the chemical composition, bond connectivity, and molecular topology of molecules. This approach places a stronger emphasis on molecular topology and chemical information, providing a representation of the molecule's overall structural arrangement and connectivity.

Our focus on 2D molecular generation is driven by several reasons. **Firstly**, 2D molecular representations capture essential chemical information and structural features, making them highly interpretable and suitable for various downstream applications such as virtual screening and drug design. **Secondly**, 2D molecular generation offers computational efficiency and scalability, enabling us to explore a larger chemical space and generate a higher number of diverse molecules within a reasonable time frame. **Lastly**, while 3D molecular generation is valuable for studying molecular interactions and binding modes, it often requires complex optimization techniques and is computationally more demanding. By concentrating on 2D molecular generation, we can achieve a balance between generating chemically relevant molecules and efficiently exploring chemical space for various property optimizations. We leave the incorporation of 3D conformation information into molecular design for our future work.

## B   LIMITATIONS AND POTENTIAL ISSUES

While our model, MOLGEN, achieves significant advancements in molecule generation, it is important to acknowledge some of its limitations, which open avenues for future research.

**Computational Efficiency:** The process of training and fine-tuning MOLGEN, especially with large datasets, can be computationally intensive, which may limit its usage in scenarios with limited computational resources.

**Model Interpretability:** Though MOLGEN exhibits prowess in generating molecules with designated properties and discerning vital molecular substructures, the opacity of transformer-based models complicates the understanding of the explicit rationale behind its determinations.

**Applicability Limitations:** A salient limitation of MOLGEN is its exclusive support for single-target optimization. The chemical feedback paradigm, whilst proficient in managing single-target molecular properties, may grapple with multiple targets. Disparate rankings for multiple objectives could engender ambiguity in the model's optimization trajectory, potentially culminating in less-than-optimal solutions. Future endeavors could investigate methodologies to adapt the chemical feedback paradigm to accommodate and prioritize diverse objectives.

**Generality Limitations:** In a bid to assess the versatility of MOLGEN, we extended our investigations to reaction prediction. Our fine-tuned model, devoid of any reliance on reaction templates, registered a 71.4% accuracy in predicting products from a pool of 39,990 reaction samples. While this underscores the model's capability to predict reactions to a certain degree, it's noteworthy that MOLGEN is not inherently structured for this task, thereby potentially curtailing its performance. Consequently, future research could consider designing a model architecture or training paradigm that concurrently and systematically accommodates reaction prediction, molecule generation, and other tasks.

---

[2]https://huggingface.co/zjunlp/MolGen-large
[3]https://huggingface.co/zjunlp/MolGen-large-opt
[4]https://huggingface.co/spaces/zjunlp/MolGen

## C  DATA INFORMATION

This section provides further information regarding the dataset employed in our study. The division of the molecular dataset into "synthetic" and "natural product" domains is to effectively explore and understand molecules of varying complexities and origins. The "synthetic" domain encompasses artificially synthesized chemical molecules tailored for specific needs, e.g., drug development. On the other hand, the "natural product" domain covers molecules naturally occurring, which are pivotal in biological activities and often provide insights for drug development. Natural product molecules generally exhibit greater structural complexity and diversity, often resulting from the myriad of unique chemical structures produced through natural biological processes. This classification helps us better understand the unique challenges and features of each domain.

In our research, we follow the methodologies of prior works (Polykovskiy et al., 2018) for distribution learning, where the baselines focus on synthetic molecule generation. Building upon this foundation, we have extended our scope by including the generation of natural products as a new and more challenging task. This expansion not only enhances the complexity of the tasks we address but also broadens the applicability of our model to a wider range of molecular structures encountered in various scientific domains.

For the natural product dataset, we sourced 30,926 compounds from the Natural Product Activity & Species Source Database (NPASS) [5] (Zhao et al., 2023). Out of these, we arbitrarily chose 30,126 molecules for training and reserved 800 molecules for testing, utilizing the same sets for all ensuing molecule generation tasks.

The characteristics of our datasets are depicted in Appendix Table 1. It is apparent that the natural product dataset manifests a distinctive distribution in comparison to the synthetic dataset, characterized by a broader spectrum of p-logP scores and reduced QED scores. This underscores the augmented complexity intrinsic to the optimization of natural product properties.

Appendix Table 1: Data statistics.

| DATASET | LENGTH | | | PENALIZED LOGP | | | QED | | |
|---|---|---|---|---|---|---|---|---|---|
| | MIN | MAX | MEAN | MIN | MAX | MEAN | MIN | MAX | MEAN |
| MOSES | 13 | 55 | 35 | -10.241 | 3.329 | -0.027 | 0.191 | 0.948 | 0.807 |
| ZINC250K | 8 | 72 | 37 | -22.189 | 5.073 | -0.622 | 0.117 | 0.948 | 0.732 |
| NATURAL PRODUCT | 2 | 436 | 55 | -51.083 | 17.691 | -2.186 | 0.005 | 0.944 | 0.438 |

## D  RELATED WORK

### D.1  DEEP GENERATIVE MODELS

In the last decade, significant strides have been made in the field of deep molecule generation (Gómez-Bombarelli et al., 2018). An array of molecular graph-based generative methods have surfaced (Ma et al., 2018; Simonovsky & Komodakis, 2018; Jin et al., 2020; Zang & Wang, 2020; Luo et al., 2021), while another branch has treated this task as a sequence generation problem with a preference for SMILES Kusner et al. (2017); Gómez-Bombarelli et al. (2018); Segler et al. (2018); Kwon et al. (2021). Based on them, existing approaches can be broadly categorized into four venues. **Bayesian Optimization** (Gómez-Bombarelli et al., 2018; Jin et al., 2018; Winter et al., 2019) learns a continuous latent space of molecules and optimizes the target properties by navigating through this space, but it often demands a protracted evaluation time to optimize the objective function (Du et al., 2022b). **Reinforcement Learning** approaches utilize an agent to select actions (e.g., adding substructures) in an explicit chemical space to enhance desired properties (Cao & Kipf, 2018; Popova et al., 2018; You et al., 2018; Popova et al., 2019; Shi et al., 2020b; Zang & Wang, 2020). However, these methods can suffer from high variance (Xie et al., 2021). An alternative approach is to employ a **Variational Auto-Encoder** (Simonovsky & Komodakis, 2018; Jin et al., 2019; Gómez-Bombarelli et al., 2018; Liu et al., 2018), but its performance heavily relies on the quality of the fixed-dimensional latent space. **Genetic Algorithms** (Jensen, 2019; Ahn et al., 2020; Nigam et al., 2020; Tripp & Hernández-Lobato, 2023) leverage predefined mutation and crossover rules to generate molecules.

---

[5] https://bidd.group/NPASS/

Despite their flexibility, obtaining the necessary prior knowledge and rules can be a challenge, hindering the efficiency of search process.

## D.2 PRE-TRAINED LANGUAGE MODELS

Just as the syntax of natural languages enforces a grammatical structure that facilitates the connection between words in specific ways, biological symbols also amalgamate in precise structural manners. PLMs have emerged as an intuitive solution for molecule generation, and several pioneers have already begun to harness SMILES-based language models, yielding promising performance (Bagal et al., 2022; Irwin et al., 2022; Flam-Shepherd et al., 2022; Ross et al., 2022; Chilingaryan et al., 2022; Pan, 2023). To date, the only publicly available PLM capable of tackling molecule generation tasks is Chemformer (Irwin et al., 2022), which follows BART (Lewis et al., 2020) to corrupt SMILES and optimize a reconstruction loss for pre-training. Expanding on the foundation laid by Chemformer, RetMol (Wang et al., 2023) incorporates external retrieval data to further improve the synthesis of molecules. Nonetheless, SMILES imposes and circumscribes grammatical rules, leading to a significant number of sequences within the appropriate character set not belonging to well-defined molecules. Additionally, the paucity of annotated or reference data may constrain the optimization direction of molecules. Diverging from those approaches, MOLGEN is pre-trained using SELFIES, which is immune to syntactic and semantic obstacles while permitting facile adaptation to different domains by sharing knowledge among model parameters via domain instruction. Moreover, it autonomously aligns with the objective of producing desirable molecules without the need for external annotated data.

## D.3 HALLUCINATION

In the field of Natural Language Processing (NLP), "hallucination" refers to generating text or responses that, while grammatically correct, fluent, and natural, deviate from the provided source inputs or lack factual accuracy (Maynez et al., 2020; Dziri et al., 2021; Shuster et al., 2021; Ji et al., 2023a; Rawte et al., 2023; Ye et al., 2023). Hallucinations are typically categorized into several types: Input-conflicting hallucinations (where the model's output deviates from the user's input), Context-conflicting hallucinations (where the output conflicts with information previously generated), and Fact-conflicting hallucinations (where the output contradicts established world knowledge) (Zhang et al., 2023). The causes of these hallucinations are varied, including biases in training data, the model's lack of access to real-time information, or the inherent limitations of the model in comprehending and generating contextually accurate responses. (Ji et al., 2023b; Zhang et al., 2023; Rawte et al., 2023).

The concept of "hallucination" is not restricted to the domain of NLP. Its adaptation in fields like molecular science, as seen in the term "*molecular hallucination*", reflects a similar disconnect between structural validity and functional accuracy. In this context, "*molecular hallucination*" refers to molecules generated by language models that are chemically valid but fail to exhibit desired properties or functionalities. In essence, these molecules, although structurally sound, do not meet the specific chemical criteria or functional expectations set for them, similar to how text generated by a language model might be grammatically correct but deviate from the intended message or content of the source input. This analogy aims to convey the concept of "unfulfilled potential" or "misleading outcomes" in molecular generation.

## E COMPARED BASELINES

In this section, we expound upon the baselines employed for comparison in our experiments. These baselines are reproduced using their open-source codes under identical experimental conditions. The baselines include:

- JT-VAE (Jin et al., 2018), a Variational Autoencoder (VAE)-based generative model that constructs a molecular graph by generating a scaffold junction tree and assembling its nodes.
- GCPN (You et al., 2018), a Reinforcement Learning (RL)-based method that crafts a molecule by optimizing a reward comprising adversarial loss and molecular property objectives.

- MOLGQN (Zhou et al., 2019), an RL-based approach that capitalizes on double Q-learning and chemical domain knowledge.
- MARS (Xie et al., 2021), a Markov Chain Monte Carlo sampling approach that employs an adaptive fragment-editing proposal distribution with Graph Neural Networks (GNN).
- GRAPHAF (Shi et al., 2020b), an autoregressive flow model that sequentially adds edges and nodes to generate molecular graphs.
- GRAPHDF (Luo et al., 2021), a normalizing flow model utilizing a discrete latent variable model and is fine-tuned with RL.
- LIMO (Eckmann et al., 2022), a VAE-based model leveraging a variational autoencoder-generated latent space.
- CHEMFORMER (Irwin et al., 2022), a pre-trained molecular language model operating on SMILES representations.
- RETMOL (Wang et al., 2023), a retrieval-based framework predicated on CHEMFORMER that incorporates a task-specific retrieval database to guide the generative model towards creating new molecules that fulfill the desired design criteria.
- RT (Born & Manica, 2023), a Transformer-based model pre-trained on SELFIES that generate molecules by inputting expected molecular property values along with a given molecular scaffold (with the generated molecules incorporating this scaffold), or to predict molecular property values based on an input molecule.

## F  COMPARISON WITH SMILES-BASED PLM

In this section, we delineate the disparities between two molecular language models, Chemformer (Irwin et al., 2022) and MOLGEN. For fairness, we select the large version of Chemformer for comparison in our paper, given its analogous size to MOLGEN. Both models leverage a pre-training dataset of 100 million molecules from the ZINC-15 dataset (Sterling & Irwin, 2015). MOLGEN boasts a more compact and specialized vocabulary size of 185, as opposed to Chemformer's expansive vocabulary of 523. This allows MOLGEN to more effectively encapsulate critical molecular substructure information.

Moreover, we present a more detailed discussion concerning SELFIES and SMILES.

- *Inherent Robustness*: Although chemical tools like RDKit (Landrum, 2013) can externally validate SMILES strings, the representation itself doesn't inherently ensure grammatical or chemical correctness. In contrast, the construction principle of SELFIES ensures a surjective mapping to molecular graphs.
- *Generative Capabilities*: Flam-Shepherd et al. (2022) provides further evidence by comparing the generative capabilities of deep models using SMILES and SELFIES. SELFIES consistently outperforms SMILES in validity, uniqueness, and novelty across tasks. Notably, SELFIES excels with longer and more complex molecules, whereas using SMILES becomes challenging due to increased character requirements and the heightened risk of encountering errors.
- *Quantitative Experiments*: Our paper includes quantitative experiment outcomes. Table 1 and Figure 6 encompass comparative analyses of SMILES and SELFIES from distribution learning and molecule generation perspectives. Note that the MolGen version in this comparison does not use the chemical feedback mechanism.
- *About SMILES*: We respect and recognize SMILES's significant contributions as a molecular descriptor. Our inclination towards SELFIES is motivated by its inherent validity in molecular generation and its simpler vocabulary, ideal for molecular language pretraining.

## G  EXPERIMENT DETAILS AND METRICS

In this section, we elucidate the evaluation metrics, training procedures, and hyperparameters utilized for each task and dataset within our experiments. MOLGEN is implemented using Pytorch and

trained on 6 Nvidia V100 GPUs. The specific experimental settings and parameters are presented in Appendix Table 2.

Appendix Table 2: Hyper-parameter settings.

| HYPER-PARAMETERS | VALUE |
|---|---|
| maximum sequence length | {55, 148, 436} |
| learning rate | {1e-5, 3e-5, 1e-4} |
| batch size | {8, 32, 64, 200, 256} |
| weight of rank loss $\alpha$ | {1,3,5} |
| prefix length | 5 |

### G.1 TWO-STAGE PRE-TRAINING

In the first stage of pre-training, we train a Seq2seq model to learn the structure, grammar, and intrinsic semantic information of SELFIES. To efficiently share parameters and knowledge, during the second stage of pre-training, we train the domain-agnostic molecular prefixes across two molecular domains. It is noteworthy that the pre-training objectives in both the first and second stages are aligned. Subsequently, we initialize the prefixes for each task with the pre-trained prefixes and optimize them for that particular task.

We utilize the LAMB optimizer, employing a linear warm-up of the learning rate for the first 180,000 gradient updates, succeeded by a linear decay for the remaining training steps. This process comprised 600 million steps with a batch size of 256 molecules per GPU.

### G.2 MOLECULAR DISTRIBUTION LEARNING

We outline the metrics employed to evaluate the performance of the generative models in our experiments, encompassing:

- *Validity*, which gauges the proportion of generated molecules adhering to valence rules.

- *Fragment similarity (Frag)*, comparing the distribution of BRICS fragments in the generated and reference sets. For instance, the Frag metric will be high if the molecules in both sets share similar fragments. Conversely, if some fragments are over- or under-represented (or entirely absent) in the generated set, the metric will be low.

- *Scaffold similarity (Scaff)* comparing the frequencies of Bemis–Murcko scaffolds (comprising all molecule's linker fragments connecting rings and ring structures) in the generated and reference sets. Specifically, if the model seldom produces a specific chemotype from a reference set, the metric will be low.

- *Similarity to the nearest neighbor (SNN)*, which measures the average Tanimoto similarity between a molecule from the generated set and its nearest neighbor molecule in the reference dataset. If the generated molecules deviate significantly from the manifold of the reference set, then the similarity to the nearest neighbor will be low.

- *Internal diversity (IntDiv)*, assessing the chemical diversity of generated molecules by calculating the average Tanimoto coefficient within the generated set.

- *Fréchet ChemNet Distance (FCD)*, considering chemically and biologically pertinent information about molecules. It can discern if the generated molecules share similar biological and chemical properties with real molecules.

- *Novelty*, measuring the percentage of the generated molecules that are not present in the training set and assessing the ability to explore the unknown chemical space.

To obtain the results detailed in Table 1, MOLGEN is trained using the AdamW optimizer with a batch size of 200 for the MOSES dataset and 32 for the natural product dataset on 6 Nvidia V100 GPUs for 100 epochs. A linear warm-up of 20000 steps was also employed.

### G.3 TARGETED MOLECULE DISCOVERY & CONSTRAINED MOLECULAR OPTIMIZATION

#### G.3.1 SIMPLE PROPERTIES

We utilize properties such as p-logP and QED as they are commonly used benchmarks in the field.

- *P-logP* refers to the logP score penalized by ring size and synthetic accessibility.
- *QED* estimates the drug-likeness of a molecule quantitatively.

For computation, p-logP and QED scores are calculated by empirical prediction models, and we employ the script based on the official implementation (Shi et al., 2020b) for comparability.

#### G.3.2 PROTEIN TARGETS

Binding affinity pertains to the strength of the interaction between a drug-like molecule and its target protein. We focus on optimizing binding affinity for two human proteins:

- *Human Estrogen Receptor (ESR1)*: A well-studied protein targeted by drugs for breast cancer treatment. This choice is driven by its clinical relevance and the availability of numerous known binding molecules, enabling effective comparison with generated compounds. MolGen utilizes solely the crystal structure of the protein (PDB 1ERR) for docking calculations and binding site information, without access to additional data on known binders.
- *Human Peroxisomal Acetyl-CoA Acyl Transferase 1 (ACAA1)*: Despite lacking known binders, this enzyme possesses a crystal structure (PDB 2IIK) featuring a potential drug-binding pocket. Identified via the Structural Genomics Consortium, this protein is recognized as a potentially disease-relevant target, possessing a documented crystal structure but devoid of known binding molecules.

The determination of binding affinity employs AutoDockGPU (Santos-Martins et al., 2021).

For both targeted molecule discovery and constrained molecular optimization tasks, we employ the chemical feedback paradigm to align the PLM with the optimization objectives. Initially, we use the pre-trained MOLGEN to generate 30 candidates for each data sample in synthetic compounds and 8 candidates for natural products. We then train the model on 6 Nvidia V100 GPUs for 100 epochs. The batch size is set to 6 for both the synthetic and natural product datasets. We utilize the AdamW optimizer, incorporating a linear warm-up of 20,000 steps.

## H ADDITIONAL VISUALIZATION OF MOLECULES GENERATED BY MOLGEN

### H.1 MOLECULAR DISTRIBUTION LEARNING

In this section, we furnish additional visual insights underscoring the prowess of MOLGEN in the realm of distribution learning. Appendix Figure 1 offers a comparative view of molecules from the training set and those generated by MOLGEN for both natural product and synthetic datasets.

The illustrated molecules provide a visual representation of how effectively MOLGEN is able to capture and reproduce the structural characteristics of molecules from different domains. This is particularly noteworthy given the substantial structural variation between molecules in the natural product and synthetic datasets. The ability of MOLGEN to generate molecules that so closely resemble the training set highlights its capability to learn and reproduce the underlying distribution of molecular structures across diverse chemical domains.

### H.2 TARGETED MOLECULE DISCOVERY

In this section, we present additional visualizations to further substantiate our claims made in the main text. To provide a more nuanced understanding of the changes in molecular properties depicted in Figure 7, we illustrate the distribution dynamics of the p-logP score in Appendix Figure 2. This reaffirms that our pre-trained MOLGEN model effectively learns the distribution of molecular

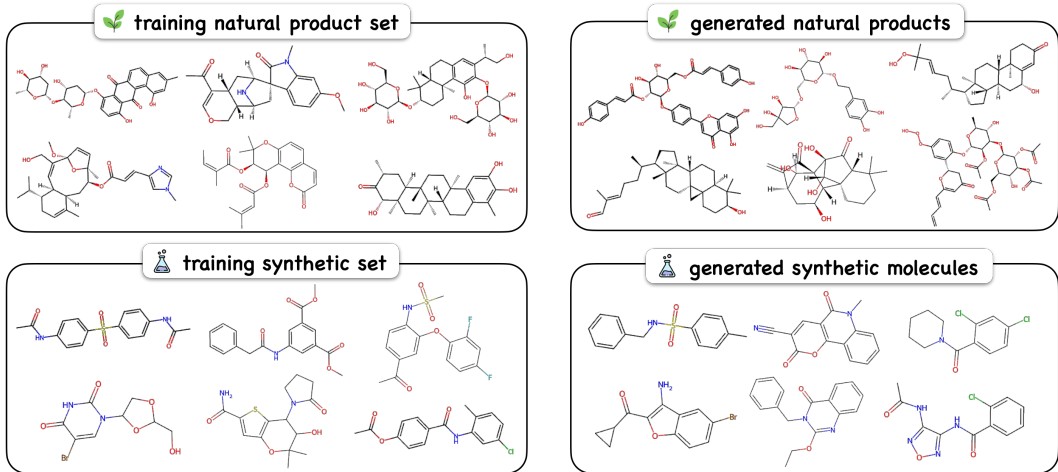

Appendix Figure 1: Comparison of visualizations of training and generated molecules.

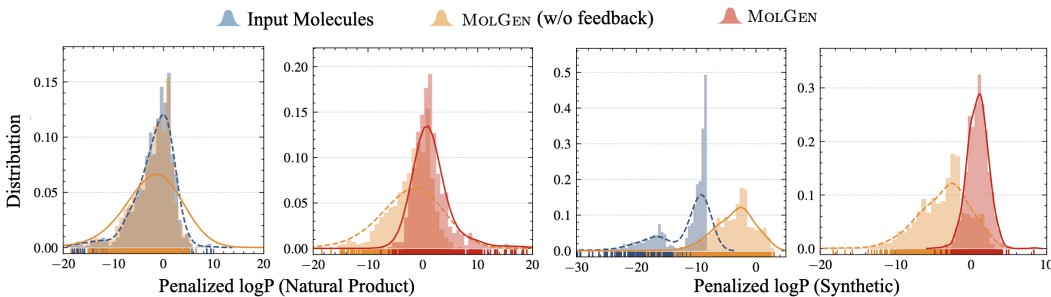

Appendix Figure 2: Property variations across different MOLGEN configurations.

properties, and that the chemical feedback paradigm enhances the property scores of the generated molecules, aligning them closer to the desired attributes.

Moreover, we display the molecules with the highest scores from both data sources in Appendix Figure 3. From this, we can deduce that although ultra-long carbon chains are frequently observed in molecules with high p-logP scores, MOLGEN is capable of maintaining high scores while preserving structural diversity. Furthermore, MOLGEN is adept at generating molecules with high QED scores while retaining the structural features characteristic of various domains.

## H.3 CONSTRAINED MOLECULAR OPTIMIZATION

In Appendix Figure 4, we provide more illustrations of constrained optimization examples for both QED and p-logP scores. These examples further highlight the proficiency of MOLGEN in optimizing molecular properties while maintaining their fundamental structures. Moreover, MOLGEN demonstrates remarkable performance even in more challenging tasks of optimizing natural products, underlining its exceptional ability to navigate and explore a broader chemical space.

## H.4 MORE MOLECULE SAMPLES OF ATTENTION VISUALIZATION

Lastly, we evaluate the representation capabilities of different PLMs by visualizing the attention weights of each token within an identical molecule, using the same setting as shown in Figure 8.

As depicted in Appendix Figure 5, MOLGEN allocates more attention to chemically significant functional groups like *carboxamide*, which demands high energy to break, and *carboxyl*, which exhibits strong polarity. In contrast, the attention mechanism in SMILES-based PLM tends to scatter across less relevant tokens, thereby diluting its focus. This demonstrates the advantage of the fine-

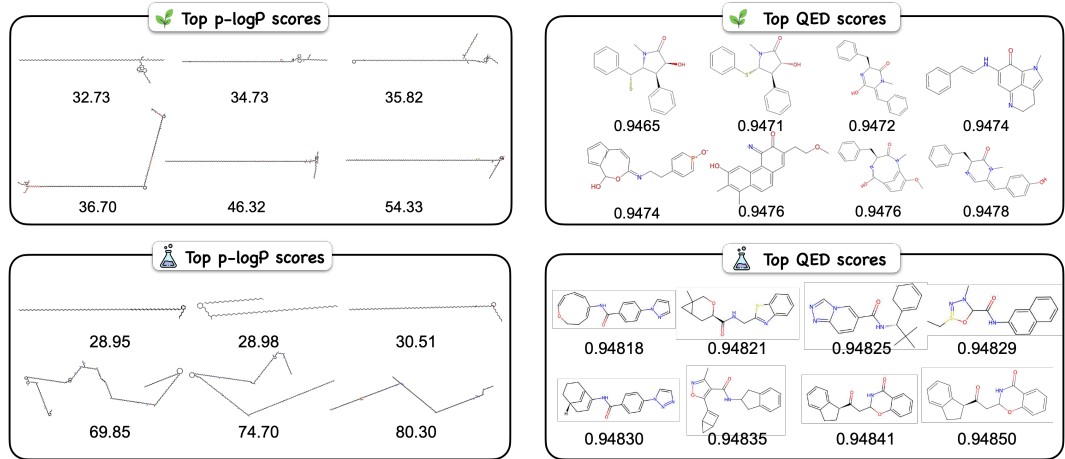

Appendix Figure 3: Samples of molecules with high penalized logP and QED scores generated by MOLGEN in natural products (left) and synthetic (right) domains.

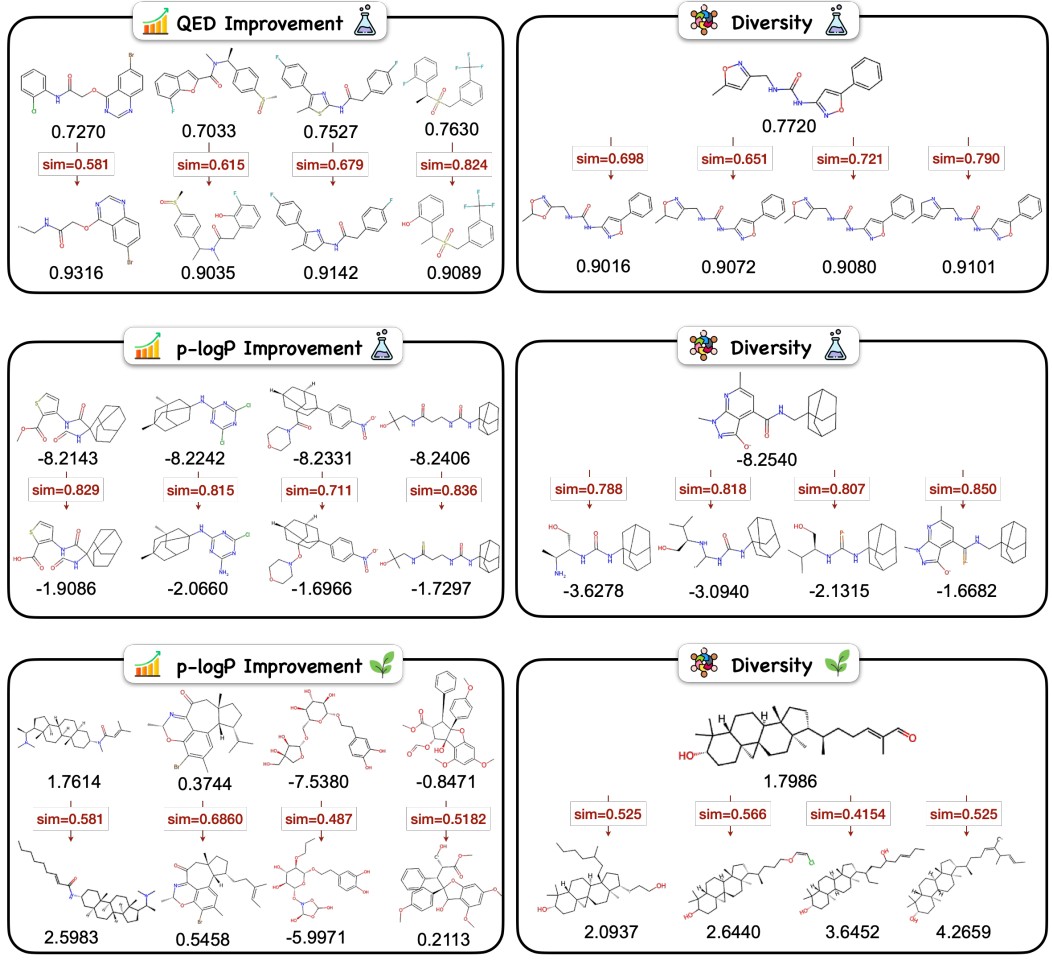

Appendix Figure 4: Further illustrations of constrained optimization using MOLGEN. Left: Examples of constrained molecular optimization for synthetic molecules (top two rows) and natural products (bottom row). Right: Optimized molecules derived from the same starting molecule, showcasing the diversity of outputs.

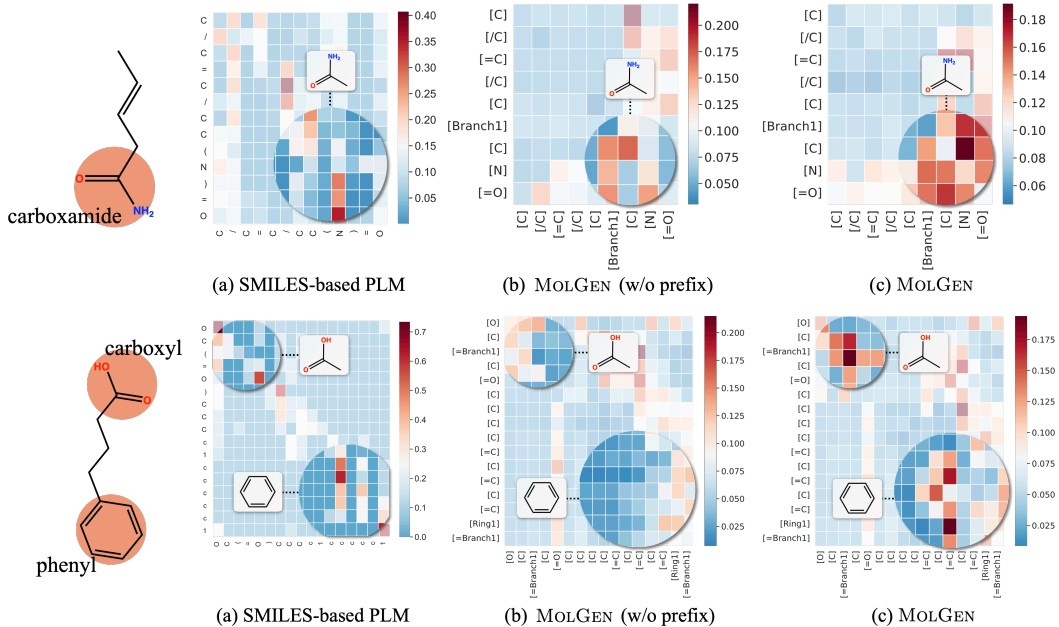

Appendix Figure 5: Visualization of the learned attention map.

grained vocabulary of MOLGEN, which can accurately identify and concentrate on pivotal structural components within molecules.

Furthermore, when we compare MOLGEN with MOLGEN (w/o prefix), we observe that the former exhibits a more focused attention pattern. It is more adept at homing in on chemically meaningful substructures and avoids unnecessary dispersion of attention. This suggests that the incorporation of domain-agnostic molecular prefixes in MOLGEN effectively guides the model's attention towards regions of significance in the molecules, thus enhancing its ability to discern vital chemical structures.

## H.5 MORE ABLATION STUDIES

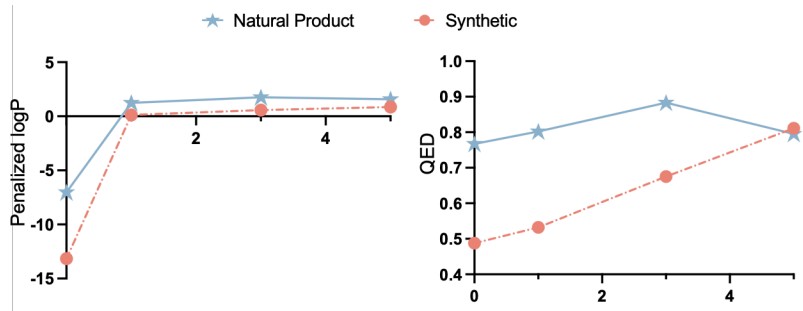

Appendix Figure 6: The impact of the hyperparameter $\alpha$.

We then explore the impact of the hyperparameter $\alpha$. As illustrated in Figure 6, an $\alpha$ value of 0 indicates no chemical feedback. When $\alpha$ is increased to 3, the model demonstrates a superior ability to optimize molecules compared to an $\alpha$ of 1. However, increasing $\alpha$ to 5 does not necessarily lead to better performance than at an $\alpha$ of 3. During actual operation, it is necessary to adjust parameters according to specific requirements. Based on experience, setting $\alpha$ to either 3 or 5 is recommended.

Additionally, we investigate the impact of label smoothing on the diversity of molecules generated by the model, employing IntDiv as the metric. IntDiv assesses the chemical diversity of generated molecules by calculating the average Tanimoto coefficient within the generated set.

As shown in Appendix Figure 7, the model with label smoothing does not overly rely on singular, frequently occurring patterns learned from the training data. Consequently, this enhances the diversity and creativity of the molecules generated.

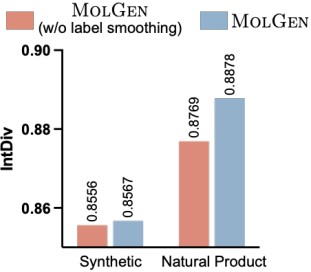

Appendix Figure 7: The impact of label smoothing.

Appendix Table 3: The impact of prefix tuning.

| MODEL | IMPROVEMENT | |
| --- | --- | --- |
| | $\delta = 0.6$ | $\delta = 0.4$ |
| MOLGEN (w/o prefix) | 11.63 (0.18) | 10.23 (1.47) |
| MOLGEN | ↑0.45 12.08 (0.82) | ↑2.12 12.35 (1.21) |

To investigate the impact of prefix tuning on the model, we present the mean (and standard deviation) of penalized logP improvement for molecules generated compared to inputs under varying similarity constraints, as detailed in Appendix Table 3. The incorporation of prefix tuning has resulted in enhanced molecule optimization performance. Taken together with Figure 8, the implementation of domain-agnostic prefix tuning not only enables the model to more effectively adapt to downstream tasks but also improves its interpretability.

