# OpenReview forum: "Domain-Agnostic Molecular Generation with Chemical Feedback"
_ICLR.cc/2024/Conference — ICLR 2024 poster_

### Official Review · Reviewer_13gf · 2023-10-30

**Soundness:** 3 good
**Presentation:** 3 good
**Contribution:** 3 good
**Rating:** 8
**Confidence:** 4

**Summary:**

This paper proposes a pre-trained molecular language model called MOLGEN to generate molecules from SELFIES with self-feedback. Specifically, MOLGEN contains two pre-training stages: 1) molecular language syntax learning using the BART model; and 2) domain-agnostic molecular prefix tuning using two sets of tunable prefix vectors in multi-head attention layers. Then, the authors use a self-feedback paradigm to align the pre-training language model with the anticipated chemical preferences in the downstream phase (i.e., the authors align the model’s probabilistic rankings of diverse molecular responses with preference rankings observed in actual chemical contexts).

**Strengths:**

Strengths:

- This paper is well organized and written.

- A two-stage domain-agnostic molecular pre-training model based on BART with SELFIES is proposed. Then, a self-feedback paradigm is used to alleviate the molecular hallucination problem.

**Weaknesses:**

Weaknesses:
- The experiment is incomplete. I suggest that the authors conduct more comprehensive experiments (i.e., baseline comparisons and ablation studies) to demonstrate the effectiveness of the proposed MOLGEN model. For example,
    - $\bf baseline~comparison$: most of the baseline models in the experiment are from before 2022 (e.g., JT-VAE (2018), GCPN (2018), GraphAF (2020), and GraphDF (2021)). I suggest the authors to compare more latest baselines to validate the MOLGEN.
    - $\bf ablation~study$: the authors only performed ablation study on the self-feedback mechanism. In Eq. 8, the authors used the "soft" label to smooth the one-hot distribution (Eq. 2) into the target distribution. This is one of the contributions of the paper. The effectiveness of the "soft" label also needs to be verified in the ablation study. Also, the hyper-parameter "$\alpha$" controls the balance of the two losses and has an important impact on the performance of the model. How to choose a and the effect of different $\alpha$ on the model performance should also be demonstrated in the experiments.

- The description of the model (Section 2) is not clear. More details need to be presented. For example,
    - Does the "$l$" in the paragraph above Eq. (1) indicate the total number of SELFIES in the dataset? Is it the same as the "$l$" in "two sets of "$l$" tunable prefix vectors" above  Eq. (3)?
    - What do the two tunable prefix vectors refer to, and why are these two vectors used in multi-head? How to prove that the performance is improved after using them?

**Questions:**

Questions:

- Please see the above.
- In the sentence "Finally, we encode the corrupted SMILES using a bidirectional model..." above Eq. (1), should it refer to "SMILES" or "SELFIES"?
- How $(S_i, S_j)$ pair in $\bf S^{*}$ are selected?

- Some typos:
    - Figures 2 and 3 have opposite locations.
    - It would be preferable to replace "ours" in tables 1, 2 and 3 with "MOLGEN". In addition, both "MOLGEN" and "MolGen" appear in the context, which should be consistent.
    - The masking and start tokens are denoted by "[MASK]" and "[S]" in the context, but are "[Mask]" and "[s]" in Figure 3. Consistency is needed.

---

> ### Author Response · Authors · 2023-11-17
>
> We sincerely thank you for your insightful feedback. We have addressed your concerns below and hope our responses provide clarity:
>
> **1. More baselines**
>
> Our selection of baseline models included notable ones like LIMO, Chemformer, and RetMol from post-2022, chosen for their well-recognized effectiveness in this field. It's important to note that finding universally recognized and effective baseline models in this rapidly evolving field can be challenging, especially with some newer models lacking publicly available code or comprehensive documentation. According to your recommendation, we have now incorporated a new baseline model, RT[1], into our analysis. **(Highlighted in Table 4 Page 7, and Appendix E Page 18)**
>
> **2. More ablation studies**
>
> In response to your suggestions, we have added the results of these ablation studies to **Appendix H.5 (Highlighted in Page 23)**.
>
> - **Effect of label smoothing**
>
> We investigate the impact of label smoothing on the diversity of molecules generated by the model, employing IntDiv as the metric. IntDiv assesses the chemical diversity of generated molecules by calculating the average Tanimoto coefficient within the generated set. As shown in the following table, the model with label smoothing does not overly rely on singular, frequently occurring patterns learned from the training data. Consequently, this enhances the diversity and creativity of the molecules generated.
>
> |Model|Synthetic|Natural Product|
> |-|-|-|
> |MolGen (w/o label smoothing)|.8556|.8769|
> |MolGen|.8567 **(+0.12%)**|.8878 **(+1.24%)**|
>
> - **Effect of $\alpha$ parameter**
>
> We also explore the impact of the hyperparameter $\alpha$. As illustrated in the following tables, an $\alpha$ value of 0 indicates no self-feedback.  When $\alpha$ is increased to 3, the model demonstrates a superior ability to optimize molecules compared to an $\alpha$ of 1. However, increasing $\alpha$ to 5 does not necessarily lead to better performance than at an $\alpha$ of 3. During actual operation, it is necessary to adjust parameters according to specific requirements. Based on experience, setting $\alpha$ to either 3 or 5 is recommended.
>
> |P-logP|Synthetic|Natural Product|
> |-|-|-|
> |$\alpha$=0|-13.152|-7.031|
> |$\alpha$=1|0.128|1.235|
> |$\alpha$=3|0.569|1.762|
> |$\alpha$=5|0.860|1.569|
>
> |QED|Synthetic|Natural Product|
> |-|-|-|
> |$\alpha$=0|0.488|0.767|
> |$\alpha$=1|0.532|0.802|
> |$\alpha$=3|0.675|0.883|
> |$\alpha$=5|0.811|0.795|

---

> > ### Author Response · Authors · 2023-11-17
> >
> > **3.  Clarify in Section 2**
> >
> > We apologize for any confusion caused by the notation in our manuscript.
> >
> > - **The use of $l$:** To clarify, in Eq. (1), $l$ refers to the length of the SELFIES strings in the dataset. In contrast, the $l$ mentioned in the context of "two sets of $l$ tunable prefix vectors" above Eq. (3) actually refers to the length of the prefix vectors themselves. We have revised the manuscript, changing this latter instance to $m$ for clearer differentiation. **(Highlighted in Section 2.1, Page 3, Last paragrapgh)**
> >
> > - **Purpose of prefix vectors:** The two tunable prefix vectors, $\boldsymbol{P}_k$ (prefix for keys) and $\boldsymbol{P}_v$ (prefix for values), serve as additional context that is prepended to the keys and values in each attention head. These prefixes allow the model to adapt its attention mechanism to capture broader, domain-level patterns that are not immediately apparent from the input sequences alone.
> >
> > - **Usage in multi-head attention:** In a multi-head attention mechanism, different heads can focus on different aspects of the input. The incorporation of these prefix vectors across all heads ensures that each aspect of attention computation, regardless of the focus of a particular head, is informed by the same baseline domain knowledge. This consistency is crucial for ensuring that the model's learning is not just specific to the nuances of the input data but is also aligned with broader domain-specific patterns.
> >
> > - **Prove the performance:**
> >
> > In response to your valuable suggestion, we have now incorporated an ablation study in our manuscript **(Highlighted in Appendix H.5, Page 24)**. To investigate the impact of prefix tuning on the model, we present the mean (and standard deviation) of penalized logP improvement for molecules generated compared to inputs under varying similarity constraints, as detailed in the following table.  This data demonstrates that prefix tuning enhances the model's performance in optimizing molecules.
> >
> > |Method|sim=0.6|sim=0.4|
> > |-|-|-|
> > |MolGen (w/o prefix)|11.63±0.18|10.23±1.47|
> > |MolGen|12.08±0.82 **(+0.45)**|12.35±1.21 **(+2.12)**|
> >
> > Additionally, **Figure 8 (Page 9)** provides a comparison of the model's attention to key substructures, both with and without prefix tuning. For enhanced clarity, we have also included a tabular representation of the data visualized in this figure, offering a more direct interpretation of the impact of prefix tuning.
> >
> > |Model|SAL|
> > |-|-|
> > |SMILES-based PLM|0.396±0.054|
> > |MolGen(w/o prefix)|0.476±0.095 **(+20.2%)**|
> > |MolGen|0.495±0.105 **(+25.0%)**|
> >
> > As indicated in the aforementioned tables, the implementation of domain-agnostic prefix tuning not only enables the model to more effectively adapt to downstream tasks but also improves its interpretability.
> >
> > **4. Pair selected method**
> >
> > In the candidate SELFIES set $\mathcal{S^*}$, every distinct pair of SELFIES forms a ($S_i$, $S_j$) pair. This means that for any two different SELFIES strings in $\mathcal{S^*}$, say $S_i$ and $S_j$, we consider them together as a pair for our analysis.
> >
> > **5. Other typos**
> >
> > We are grateful for your attention to these details and pointing out these oversights. We have made the following revisions to our paper according to your suggestions:
> >
> > - Corrected "SMILES" to "SELFIES" **(Highlighted in Section 2.1, Page 3)**.
> > - Swapped the positions of Figure 2 and 3 **(Page 3)**.
> > - Replaced the term "ours" with our model's specific name, "MOLGEN" **(Table 1-3)**.
> > - Ensured that the name of our model is consistently spelled as "MOLGEN" in all figures and context **(Figure 6-8, and Appendix Figure 2, 5)**.
> > - Revised "[Mask]" to "[MASK]" in Figure 2 **(Page 3)**.
> >
> > Thank you once more for your invaluable suggestions, which have significantly helped in enhancing the rigor of our experiments and the completeness of our paper!
> >
> > [1] Regression Transformer enables concurrent sequence regression and generation for molecular language modelling. In NMI 2023.

---

> > > ### Comment · Reviewer_13gf · 2023-11-22
> > > **After rebuttal**
> > >
> > > Thanks for the effort the authors made to clarify and improve the manuscript.
> > > I would like to rise my score.

---

> > > > ### Author Response · Authors · 2023-11-22
> > > > **Thanks!**
> > > >
> > > > Thank you immensely for your feedback! We are gratified to know that we have successfully addressed your concerns.

---

### Official Review · Reviewer_Hnvb · 2023-10-30

**Soundness:** 2 fair
**Presentation:** 1 poor
**Contribution:** 2 fair
**Rating:** 6
**Confidence:** 4

**Summary:**

This paper introduces MolGen, a molecule language model with 700M parameters built on the Bart architecture. Unlike its predecessor, Chemformer, MolGen employs SELFIES representations for molecules, ensuring 100% validity in molecule generation. Additionally, the paper presents a fine-tuning technique called 'self-feedback,' which aligns MolGen's generation probabilities with the chemical properties of the generated molecules.

The proposed method has shown significant improvement over previous methods. However, I have some major concerns on the paper's presentation, see the weakness section for details.

**Strengths:**

* The model shows significant improvements in the compared benchmarks, including distribution learning, targeted molecule discovery and constrained optimization.
* The proposed pretraining method is simple and plausible.

**Weaknesses:**

The submission contains significant issues, including the misuse of two major concepts and some questionable claims. Given the centrality of these concepts to the paper's overall argument, I recommend that the paper is not suitable for publication in its current form.

* **Concept Misuse: Molecular Hallucination:** The authors introduce the notion of "molecular hallucination" as the generation of molecules that do not exhibt the desired properties or follow human preferences. The term appears to be a misnomer, as it does not align with the traditional understanding of "hallucination" in the NLP domain. In NLP literature, hallucination generally refers to the generation of fictitious or nonexistent entities or events [1,2,3]. Conversely, the authors assert that their "hallucinated" molecules are chemically valid, meaning they are neither fictitious nor nonexistent. Therefore, the use of the term "hallucination" could be misleading and should be reconsidered for clarity and conceptual consistency.
* The related works on hallucination are missing.
* **Concept Misuse: Self-feedback:** The authors refer to their fine-tuning approach as the "self-feedback paradigm." However, the proposed method is more like "**external-feedback**", instead of "self-feedback". In short, the proposed fine-tuning method is to align the molecule generation probabilities to chemical property scores **that are measured by an external model**. Given this external dependency, the term "external-feedback paradigm" would be a more accurate and descriptive name for the fine-tuning method.
* **Missing Ablation Study:** MolGen is pretrained in two phases. In the second phase, MolGen is adapted on two datasets using a method called domain-agnostic molecular prefix tuning. After reading the submission, it remain unclear to me how the second phase can help other downstream tasks. Moreover, the ablation study on the second pretraining phase is missing. It is also unclear how is the proposed prefix tuning method compared against other parameter efficient tuning methods, like LoRA?
* **Unclear Figures:**
  * What are the x-axis of the two subfigures on the right of Figure 7?
  * What are the x-axis and y-axis of the two subfigures on the right of Figure 8?
  * What are the x-axis of the Appendix Figure 2?

* **Novelty:** This is not the first molecule language model pretrained using molecule's SELFIE representations. [4] is also pretrained using SELFIES. The relevance and diffierence to [4] should be discussed.
* **Dubious claims:**
  * In the second paragraph of introduction, the authors argue that previous works `are limited by their heavy reliance on task-specific tuning`. However, the proposed method also relies on a fine-tuning stage for downstream tasks.
  * In the third paragraph of introduction, the authors claims that `almost all previous studies have focused primarily on synthetic molecules, neglecting natural products`. However, the literature review to support this claim is missing.
  * The opening paragraph of Section 2.2 describes a scenario where a molecule has one substructure that is effective for a specific task but is counteracted by another, ineffective substructure. However, it remains unclear how the proposed method addresses this issue.
  * In the second paragraph of the introduction, the authors claim that `the brittleness of SMILES may lead to a high proportion of generated chemically invalid strings`. Are there any citation to support this claim?








**Reference:**

[1] On Faithfulness and Factuality in Abstractive Summarization. In ACL 2020.

[2] Neural Path Hunter: Reducing Hallucination in Dialogue Systems via Path Grounding. In EMNLP 2021.

[3] Retrieval Augmentation Reduces Hallucination in Conversation. In EMNLP Findings 2021

[4] Regression Transformer enables concurrent sequence regression and generation for molecular language modelling. In NMI 2023.

**Questions:**

* Is the synthetic dataset in Section 3.1 referring to ZINC250K? If it is, directly use the name ZINC250k can improve the clarity.

---

> ### Author Response · Authors · 2023-11-17
>
> We sincerely thank you for your insightful feedback. We have addressed your concerns below and hope our responses provide clarity:
>
> **1. Concept of molecular hallucination**
>
> We apologize for any ambiguity in our expression, but we wish to clarify that our concept of "molecular hallucination" is indeed aligned with the NLP concept of hallucination. In NLP, hallucination refers to generating text or responses that, while grammatically correct, fluent, and natural, deviate from the provided source inputs (faithfulness) or lack factual accuracy (factualness) [1] [2] [3] [4].
>
> Therefore, in our study, "molecular hallucination" is used to describe the phenomenon where molecules created by language models, although chemically valid and adhering to basic structural rules (similar to producing grammatically correct sentences), fail to possess the anticipated chemical properties (akin to producing text that deviates from the intended message or content of the source input). This concept is consistent with the understanding of hallucination in NLP, and as you may have noticed, other reviewers have also concurred with this viewpoint.
>
> To enhance clarity, we have included related works about hallucination and further clarified the concept of "molecular hallucination" **(Highlighted in Appendix D.3)**.
>
> **2. Concept of self-feedback**
>
> We chose the term "self-feedback" to underscore the model's self-driven learning and optimization process. In this process, the model guides its own learning and optimization based on the evaluation results of molecules it autonomously generates, though these evaluations are aided by an external scoring function. This term is specifically chosen to differentiate our work from studies that use responses from other models as their feedback signal sources [5]. In contrast, our feedback mechanism is based on the responses generated by our own model.
>
> In response to your feedback, we have revised our wording to better convey this concept to readers, emphasizing the model's self-reflective learning cycle, despite incorporating external scoring **(Highlighted in Section 2.2, Page 5)**.
>
> **3. Additional ablation study**
>
> We apologize for any confusion that may have arisen from our presentation and would like to clarify that the domain-agnostic molecular prefix-tuning in the second phase is distinct from parameter-efficient tuning methods. As mentioned in Page 3, the penultimate paragraph, this approach impacts the entire model's parameters,  boosting its ability to understand and adapt across various molecular domains.
>
> In response to your valuable suggestion, we have now incorporated an ablation study in our manuscript **(Highlighted in Appendix H.5, Page 24)**. To investigate the impact of prefix tuning on the model, we present the mean (and standard deviation) of penalized logP improvement for molecules generated compared to inputs under varying similarity constraints, as detailed in the following table.  This data demonstrates that prefix tuning enhances the model's performance in optimizing molecules.
>
> |Method|sim=0.6|sim=0.4|
> |-|-|-|
> |MolGen (w/o prefix)|11.63±0.18|10.23±1.47|
> |MolGen|12.08±0.82 **(+0.45)**|12.35±1.21 **(+2.12)**|
>
> Additionally, **Figure 8 (Page 9)** provides a comparison of the model's attention to key substructures, both with and without prefix tuning. For enhanced clarity, we have also included a tabular representation of the data visualized in this figure, offering a more direct interpretation of the impact of prefix tuning.
>
> |Model|SAL|
> |-|-|
> |SMILES-based PLM|0.396±0.054|
> |MolGen(w/o prefix)|0.476±0.095 **(+20.2%)**|
> |MolGen|0.495±0.105 **(+25.0%)**|
>
> As indicated in the aforementioned tables, the implementation of domain-agnostic prefix tuning not only enables the model to more effectively adapt to downstream tasks but also improves its interpretability.
>
> **4. Modifications to the figures**
>
> We have revised these figures to better facilitate reader understanding.
>
> - For **Figure 7 (Highlighted in Page 8)**, the x-axis represents the QED values, and the y-axis shows their distribution.
> - For **Figure 8 (Highlighted in Page 9)**, the first subfigure has the x-axis representing the Substructure Attention Level (SAL) values, while the second subfigure's x-axis also represents SAL values and the y-axis shows their distribution.
> - For **Appendix Figure 2 (Highlighted in Page 21)**, the x-axis represents the property values and the y-axis shows their distribution.

---

> ### Author Response · Authors · 2023-11-17
>
> **5. Related paper**
>
> Thank you for bringing this to our attention. While it is true that this study [6] also uses molecule's SELFIES representations for pretraining, the focus and methodology differ significantly from our work. In the referenced paper, the approach reinterprets regression as a sequence modeling task, where training combines property tokens with text tokens. Their model is engineered to generate molecules by inputting expected molecular property values along with a given molecular scaffold (with the generated molecules incorporating this scaffold), or to predict molecular property values based on an input molecule.
>
> In contrast, our model is specifically designed to generate entirely new molecules based on existing ones, without being confined to a predetermined scaffold. This key difference in our objectives and techniques distinctly sets our research apart from the approach outlined in the cited study.
>
> Following your suggestion, we have now included the referenced paper as a new baseline in **Table 4 (Highlighted in Page 7)** and a discussion of this study in **Appendix E (Highlighted in Page 18)** of our revised manuscripts.
>
> **6. Other revisions**
>
> We are grateful for your attention to these details and pointing out these oversights. In response to your valuable suggestions, we have made the following revisions:
>
> - **Task-specific tuning:** We changed the phrase "heavy reliance on task-specific tuning" to "hard to train due to the high variance of RL". This alteration underscores our method's aim to alleviate the instability and difficulty associated with training reinforcement learning models, thereby offering a more stable and efficient process for downstream applications. **(Highlighted in Section 1, Page 1)**
>
> - **Focus on synthetic molecules:** We included additional references to support our claim about previous studies. **(Highlighted in Section 1, Page 2)**
>
> - **Examples of molecular hallucination:** To avoid any ambiguity, we have rephrased the description in the scenario where the model generates a molecule with an additional side chain. The revised text emphasizes that the perceived structural robustness is illusory and does not guarantee desirable properties, exemplifying "molecular hallucination". **(Highlighted in Section 2.2, Page 4)**
>
> - **Brittleness of SMILES:** We added necessary references to support the statement about SMILES potentially leading to meaningless sequences. **(Highlighted in Section 1, Page 2)**
>
> Thank you again for your constructive suggestions! Your insights have significantly contributed to enhancing the completeness and persuasiveness of our paper. We hope that the revisions and clarifications provided in our responses adequately address your concerns.
>
> [1] Survey of Hallucination in Natural Language Generation. In ACM Computing Surveys 2023.
> [2] A Survey of Hallucination in “Large” Foundation Models. 2023.
> [3] Siren’s Song in the AI Ocean: A Survey on Hallucination in Large Language Models. 2023.
> [4] Cognitive Mirage: A Review of Hallucinations in Large Language Models. 2023.
> [5] Contrastive Post-training Large Language Models on Data Curriculum. 2023.
> [6] Regression Transformer enables concurrent sequence regression and generation for molecular language modelling. In NMI 2023.

---

> ### Comment · Reviewer_Hnvb · 2023-11-22
> **Followup questions**
>
> Thanks for the authors' efforts on rebuttal. I appreciate them. The response has resolved my concern on the molecule hallucination problem, and I have raised my rating accordingly. I still have the following questions about the submission.
>
> * **Regarding the concept of self-feedback.** As the authors agree, the objective function is to align the generation probability to an external model. In [1], the external model is a RLHF reward model; in this submission, the external model is the QED or plop score function. In concept, it is still external feedback but not self-feedback, because the training signal does not come from MolGen itself.
>
> * **Regarding the prefix tuning method.**  `this approach impacts the entire model's parameters`. Does it mean that the MolGen's other parameters (not the prefix) are also tuned in fine-tuning?
>
>
>
> **Reference:**
>
> [1] CONTRASTIVE POST-TRAINING LARGE LANGUAGE MODELS ON DATA CURRICULUM. 2023

---

> > ### Author Response · Authors · 2023-11-22
> >
> > We sincerely thank you for your insightful feedback. Below are our detailed responses to your concerns:
> >
> > **Regarding the concept of self-feedback:** After careful consideration of your suggestion, which also impacts the title of our paper, we have decided to follow your advice and avoid using the term "self-feedback" to prevent any misunderstandings. Specifically, we have replaced "self-feedback" with "chemical feedback" in our manuscript. Additionally, we have accordingly revised all the relevant figures in the manuscript.
> >
> > **Regarding the prefix tuning method:** Yes, you have understood correctly. As stated on Page 3 of our manuscript, both the prefix and the model parameters are tunable. This approach allows the model to effectively adapt to new tasks while maintaining proximity to the original pre-trained knowledge.
> >
> > We hope these clarifications more appropriately address your concern. Thank you once again for your valuable input in enhancing the quality of our work!

---

> > > ### Comment · Reviewer_Hnvb · 2023-11-22
> > > **Thanks for the prompt response**
> > >
> > > Thank the authors for their efforts in rebuttal. I have raised my recommendation score.

---

> > > > ### Author Response · Authors · 2023-11-22
> > > > **Thanks!**
> > > >
> > > > Thank you very much for your feedback! We are pleased that we could address your concerns.

---

### Official Review · Reviewer_A1g6 · 2023-10-31

**Soundness:** 3 good
**Presentation:** 3 good
**Contribution:** 3 good
**Rating:** 8
**Confidence:** 4

**Summary:**

This paper presents MOLGEN, a new pre-trained molecular language model dedicated to molecule generation. By reconstructing over 100 million molecular SELFIES, MOLGEN has gained in-depth knowledge of molecular structures and grammar. This understanding is amplified by the domain-agnostic molecular prefix tuning, ensuring better knowledge transferability across a wide range of domains. A crucial feature of the model is the self-feedback mechanism, which safeguards against "molecular hallucinations" by ensuring that the model's estimated probabilities align with real-world chemical propensities. Comprehensive evaluation on established benchmarks highlights MOLGEN's superior performance in properties like penalized logP, QED, and molecular docking. Further analysis confirms its adeptness in accurately capturing molecule distributions, discerning intricate structural patterns, and efficiently exploring the chemical space.

**Strengths:**

1. This paper introduces a language model designed for molecule generation, adeptly capturing deep structural and grammatical insights through the reconstruction of over 100 million molecular SELFIES.

2. The paper is well-written and easy to follow. Figures and Tables are very good.

3. Compared to previous baselines, the proposed approach showcases impressive performance. Through comprehensive experiments, the paper convincingly demonstrates that SELFIES is a superior molecular representation to SMILES for 2D molecule generation tasks. This paper also provides very insightful discussions, which can help to understand the model behaviors.

**Weaknesses:**

What potential limitations might the self-feedback mechanism introduce in molecular generation?

**Questions:**

See Weaknesses

---

> ### Author Response · Authors · 2023-11-17
>
> Thank you very much for your insightful comments. Below are our responses:
>
> - **Limitations about self-feedback**
>
> Currently, the self-feedback mechanism in our study is tailored for optimizing a single target objective. As we mentioned in **Appendix B**, this approach exhibits reduced efficacy when applied to scenarios involving multiple objectives. This is primarily due to the generation of divergent preference rankings, which leads to uncertainty in the model's decision-making process about which optimization directions to prioritize. Consequently, an area for future research could be the development of self-feedback techniques that are more adept at handling and optimizing multiple objectives simultaneously.
>
> Thank you again for your constructive suggestions! We hope our responses address your concerns.

---

### Official Review · Reviewer_Vj81 · 2023-11-07

**Soundness:** 2 fair
**Presentation:** 3 good
**Contribution:** 2 fair
**Rating:** 6
**Confidence:** 4

**Summary:**

The paper presents MolGen, a domain-agnostic molecular generation model, and its application in generating molecules using the SELFIES molecular language. The paper discusses the ability of MolGen to discern essential substructures and compares it with other molecular generation approaches using the SMILES language. The paper also introduces a self-feedback mechanism to mitigate "molecular hallucinations" and improve the generation of molecules with desired properties.

**Strengths:**

1. The paper is well-structured with clear writing and is supported by rich and lucid illustrations.

2. The application of SELFIES, as opposed to SMILES, is more concise and effective in deep generative models, facilitating the analysis of generated results.

3. The paper delves deep into the concept of 'Molecular Hallucinations' and attempts to address it, which is a beneficial discussion for the field of molecule generation.

**Weaknesses:**

1. Although an ablation study was conducted to check the self-feedback paradigm, there was no ablation experiment carried out to assess the use of SELFIES over SMILES. Instead, only a comparison was made between Chemformer and MolGen. Technically, a SELFIES-based MolecularLM implemented on BART does not seem irreplaceable. Given that many works involving molecule generation are still based on SMILES (such as MoMu, MolT5), the effectiveness of SELFIES in the work lacks further experimental verification.

2. The significance of the 'domain-agnostic molecular prefix tuning' step is questionable. It seems to be merely a measure to avoid overfitting in the overall model. Whether synthetic molecule generation and natural product generation in drug discovery can be considered two different tasks, and whether other dataset partitioning methods would have similar effects, are not explained. Therefore, the comparison of molecular distribution learning in the paper lacks persuasiveness.

**Questions:**

1. It would be highly beneficial if the authors could conduct further experiments to address the issues raised in the 'Weaknesses' section of this review.

2. Given that molecule generation typically needs to cater to a variety of requirements, have the authors considered other metrics beyond penalized logP, QED, and binding affinity for two human proteins? More persuasive experiments addressing a broader range of molecular properties could significantly enhance the applicability and robustness of the proposed model.

3. The correspondence between the attention scores and the specific molecular structures in Figure 8 and Appendix Figure 5 is not very intuitive. The current figures do not convincingly demonstrate that the SMILES-based PLM is focusing attention on less relevant positions. It would be beneficial if the authors could revise this figure to improve its clarity and interpretability, thereby aiding readers in better understanding the model's inner workings.

**Details Of Ethics Concerns:**

No.

---

> ### Author Response · Authors · 2023-11-17
>
> We sincerely thank you for your insightful feedback. Below are our detailed responses to your concerns:
>
> **1. Effectiveness of SELFIES**
>
> We apologize for any confusion that may have arisen from our presentation.
>
> - To further clarify, as detailed in Appendix F, we have conducted a comparative analysis using both SMILES and SELFIES formats. In **Table 1, Figure 6, and Figure 8**, Chemformer (S-PLM), which is pre-trained on SMILES and employs a BART architecture, serves as our baseline for comparison. This setup, with MolGen not utilizing the self-feedback mechanism, ensures an unbiased evaluation. The experimental results demonstrate that using SELFIES as the molecular language enables the generation of 100% valid molecules, while also facilitating a more focused attention mechanism in the model.
> - The decision to use SELFIES is based on its standardized and more simplified syntax, which we find to be particularly well-suited for processing by language models. This choice is reflected in our results, where SELFIES ensured the generation of 100% valid molecules, aligning well with our study's objectives. We wish to underscore that SMILES remains a vital and widely adopted format for molecular representation, and our study does not intend to diminish its importance. Instead, our utilization of SELFIES represents a novel exploration, driven by the specific objectives of our research.
>
> **2. Synthetic molecule v.s. natural product generation**
>
> The division of the molecular dataset into "synthetic" and "natural product" domains is to effectively explore and understand molecules of varying complexities and origins. The "synthetic" domain encompasses artificially synthesized chemical molecules tailored for specific needs, e.g., drug development. On the other hand, the "natural product" domain covers molecules naturally occurring, which are pivotal in biological activities and often provide insights for drug development. Natural product molecules generally exhibit greater structural complexity and diversity (further data statistics are available in Appendix C), often resulting from the myriad of unique chemical structures produced through natural biological processes. This classification helps us better understand the unique challenges and features of each domain.
>
> In our research, we follow the methodologies of prior works[1] for distribution learning, where the baselines focus on synthetic molecule generation. Building upon this foundation, we have extended our scope by including the generation of natural products as a new and more challenging task. This expansion not only enhances the complexity of the tasks we address but also broadens the applicability of our model to a wider range of molecular structures encountered in various scientific domains. Given that both synthetic and natural molecules represent a significant portion of molecules in daily life and research, we posit that our model’s proficiency in these domains indicates its capability to handle various molecular tasks effectively. In response to your suggestion, we have elaborated on this aspect in **Appendix C (Page 16)**.
>
> **3. Effectiveness of prefix tuning**
>
> We performed an additional experiment to investigate the impact of prefix tuning **(Highlighted in Appendix H.5, Page 24)**. Here, we present the mean (and standard deviation) of penalized logP improvement for molecules generated compared to inputs under varying similarity constraints, as detailed in the following table.  This data demonstrates that prefix tuning enhances the model's performance in optimizing molecules.
>
> |Method|sim=0.6|sim=0.4|
> |-|-|-|
> |MolGen (w/o prefix)|11.63±0.18|10.23±1.47|
> |MolGen|12.08±0.82 **(+0.45)**|12.35±1.21 **(+2.12)**|
>
> Additionally, **Figure 8 (Page 9)** provides a comparison of the model's attention to key substructures, both with and without prefix tuning. For enhanced clarity, we have also included a tabular representation of the data visualized in this figure, offering a more direct interpretation of the impact of prefix tuning.
>
> |Model|SAL|
> |-|-|
> |SMILES-based PLM|0.396±0.054|
> |MolGen(w/o prefix)|0.476±0.095 **(+20.2%)**|
> |MolGen|0.495±0.105 **(+25.0%)**|
>
> As indicated in the aforementioned tables, the implementation of domain-agnostic prefix tuning not only enables the model to more effectively adapt to downstream tasks but also improves its interpretability.

---

> > ### Author Response · Authors · 2023-11-17
> >
> > **4. More metrics beyond penalized logP, QED, and binding affinity**
> >
> > In our study, we select penalized logP and QED due to their widespread recognition as standard benchmarks for evaluating simple molecular properties. Additionally, we include binding affinity to assess a more complex and practical property. We believe that the combination of these metrics effectively demonstrates our method's ability to generate molecules with both simple and complex desired properties, thereby reflecting the model's versatility and generalizability.
> >
> > At this stage, while acknowledging the importance of a diverse range of metrics, we have chosen to focus on these core indicators in this work. We strongly agree with your perspective and are dedicated to identifying and integrating more practically valuable metrics in our future research endeavors, further enhancing the applicability and robustness of our model.
> >
> > **5. Improving the clarity of figures**
> >
> > In response to your comments, we have thoroughly revised both **Figure 8 and Appendix Figure 5**. These updates aim to enhance the clarity and interpretability of the correlation between attention scores and specific molecular structures.
> >
> > Thank you again for your valuable suggestions! Your input has greatly contributed to enhancing the completeness and readability of our paper.
> >
> > [1] Molecular Sets (MOSES): A Benchmarking Platform for Molecular Generation Models

---

> > ### Comment · Reviewer_Vj81 · 2023-11-23
> > **Raise Score**
> >
> > Thanks to the authors for their response. I have raised my score to 6.

---

> > > ### Author Response · Authors · 2023-11-23
> > > **Thanks!**
> > >
> > > Thank you for your valuable feedback! We are glad that our responses have successfully resolved your concerns.

---

### Author Response · Authors · 2023-11-17
**Summary of Revisions**

Dear Reviewers and AC,

We are deeply grateful for your valuable time and insightful feedback. A revised draft of our manuscript has been uploaded, with changes highlighted in magenta font for ease of reference. Below, we summarize the main revisions:

- Redraw Figure 8 (Page 9) and Appendix Figure 5 (Page 23) for more clarity and interpretability of the correlation between attention scores and specific molecular structures.
- Add necessary references in Section 1(Page 2).
- Add a new baseline and its discussion in Table 4 (Page 7) and Appendix E (Page 18).
- Add an explanation about the rationale behind classifying molecules into synthetic and natural products in Appendix C (Page 16).
- Add related work about hallucinations in Appendix D.3 (Page 17).
- Corrected the typos and unclear parts mentioned by reviewers (Pages 1, 3, 4, and 5).
- Add the ablation study about prefix tuning in Appendix Table 3 (Page 24).
To investigate the impact of prefix tuning on the model, we present the mean (and standard deviation) of penalized logP improvement for molecules generated compared to inputs under varying similarity constraints, as detailed in the following table. This data demonstrates that prefix tuning enhances the model's performance in optimizing molecules. Taken together with Figure 8 (Page 9), the implementation of domain-agnostic prefix tuning not only enables the model to more effectively adapt to downstream tasks but also improves its interpretability.
|Method|sim=0.6|sim=0.4|
|-|-|-|
|MolGen (w/o prefix)|11.63±0.18|10.23±1.47|
|MolGen|12.08±0.82 **(+0.45)**|12.35±1.21 **(+2.12)**|

- Add the ablation study about label smoothing in Appendix Figure 7 (Page 23).
We investigate the impact of label smoothing on the diversity of molecules generated by the model, employing IntDiv as the metric. IntDiv assesses the chemical diversity of generated molecules by calculating the average Tanimoto coefficient within the generated set. As shown in the following table, the model with label smoothing does not overly rely on singular, frequently occurring patterns learned from the training data. Consequently, this enhances the diversity and creativity of the molecules generated.
|Model|Synthetic|Natural Product|
|-|-|-|
|MolGen (w/o label smoothing)|.8556|.8769|
|MolGen|.8567 **(+0.12%)**|.8878 **(+1.24%)**|

- Add the ablation study about the hyperparameter in Appendix Figure 6 (Page 23).
We explore the impact of the hyperparameter $\alpha$. As illustrated in the following tables, an $\alpha$ value of 0 indicates no self-feedback.  When $\alpha$ is increased to 3, the model demonstrates a superior ability to optimize molecules compared to an $\alpha$ of 1. However, increasing $\alpha$ to 5 does not necessarily lead to better performance than at an $\alpha$ of 3. During actual operation, it is necessary to adjust parameters according to specific requirements. Based on experience, setting $\alpha$ to either 3 or 5 is recommended.
|P-logP|Synthetic|Natural Product|
|-|-|-|
|$\alpha$=0|-13.152|-7.031|
|$\alpha$=1|0.128|1.235|
|$\alpha$=3|0.569|1.762|
|$\alpha$=5|0.860|1.569|
|**QED**|**Synthetic**|**Natural Product**|
|$\alpha$=0|0.488|0.767|
|$\alpha$=1|0.532|0.802|
|$\alpha$=3|0.675|0.883|
|$\alpha$=5|0.811|0.795|

We briefly introduce the motivation, method, and contribution as follows:

Motivation:
Design a molecular language model to generate chemically valid molecules and mitigate the phenomenon of "molecular hallucinations".

Method:
- Two-stage domain-agnostic molecular pre-training.
- Self-feedback paradigm to alleviate "molecular hallucinations".

Contribution:
- The first molecular language model to generate 100% valid molecules.
- Aligning the pre-trained model with real-world chemical preferences to produce molecules with desired properties.
- Validating the model's performance on multiple optimization goals and making the code publicly available.

We sincerely hope our responses and revisions address all reviewers’ concerns.
We sincerely believe that these updates may help us better deliver the benefits of the proposed work to the ICLR community.

Thank you very much,
Authors

---

### Author Response · Authors · 2023-11-21
**Urgent Request for Re-review and Discussion**

Dear Reviewers and AC,

We genuinely value the constructive comments and insightful suggestions you provided for our work. Recognizing the approaching end of the discussion period on **November 22nd**, we kindly urge you to participate in the ongoing discussion and provide any additional insights or clarifications you may have. Your expertise is invaluable to us, and we believe your input will significantly contribute to the improvement of our work.

Thank you very much for your time and consideration. We look forward to hearing from you soon.

Authors

---

### Meta-Review · Program_Chairs · 2023-12-05

**Metareview:**

This paper proposes a domain-agnostic molecular generation model based on a SELFIES-based pre-training scheme followed by a chemical feedback algorithm with chemical preference. This approach introduces an issue of "molecular hallucination" in molecular generation, and avoids such a problem by aligning the chemical preferences and the estimated probabilities of the model. The paper thoroughly compared the proposed approach with prior works to provide a strong motivation for the research, and the experimental results are noteworthy. The authors did a good job to address the reviewers' concerns in the author-reviewer discussion period, and all reviewers support the acceptance. AC also did not find a particular weakness for rejection. Overall, AC is happy to recommend the acceptance.

**Justification For Why Not Higher Score:**

The novelty is not super.

**Justification For Why Not Lower Score:**

All reviewers suggest acceptance and AC also did not find a particular weakness for rejection.

---

### Decision · Program_Chairs · 2024-01-16

Accept (poster)